# Many-body Approximation for Non-negative Tensors

**Kazu Ghalamkari**
RIKEN AIP
kazu.ghalamkari@riken.jp

**Mahito Sugiyama**
National Institute of Informatics
SOKENDAI
mahito@nii.ac.jp

**Yoshinobu Kawahara**
Osaka University
RIKEN AIP
kawahara@ist.osaka-u.ac.jp

## Abstract

We present an alternative approach to decompose non-negative tensors, called *many-body approximation*. Traditional decomposition methods assume low-rankness in the representation, resulting in difficulties in global optimization and target rank selection. We avoid these problems by energy-based modeling of tensors, where a tensor and its mode correspond to a probability distribution and a random variable, respectively. Our model can be globally optimized in terms of the KL divergence minimization by taking the *interaction between variables (that is, modes)*, into account that can be tuned more intuitively than ranks. Furthermore, we visualize interactions between modes as *tensor networks* and reveal a nontrivial relationship between many-body approximation and low-rank approximation. We demonstrate the effectiveness of our approach in tensor completion and approximation.

## 1 Introduction

Tensors are generalizations of vectors and matrices. Data in fields such as neuroscience [8], bioinformatics [22], signal processing [7], and computer vision [30] are often stored in the form of tensors, and features are extracted from them. *Tensor decomposition* and its non-negative version [33] are popular methods to extract features by approximating tensors by the sum of products of smaller tensors. These methods usually seek to minimize the reconstruction error, the difference between an original tensor and the tensor reconstructed from obtained smaller tensors.

In tensor decomposition approaches, a *low-rank structure* is typically assumed, where a given tensor is essentially represented by a linear combination of a small number of bases. Such decomposition requires two kinds of information: structure and the number of bases used in the decomposition. The structure specifies the type of decomposition such as CP decomposition [15] and Tucker decomposition [37]. *Tensor networks*, which were initially introduced in physics, have become popular in the machine learning community recently because they help intuitive and flexible model design, including tensor train decomposition [29], tensor ring decomposition [41], and tensor tree decomposition [25]. Traditional tensor structures in physics, such as MERA [32] and PEPS [4], are also used in machine learning. The number of bases is often referred to as the rank. Larger ranks increase the capability of the model while increasing the computational cost, resulting in the tradeoff problem of rank tuning that requires a careful treatment by the user. Since these low-rank structure-based decompositions via reconstruction error minimization are non-convex, which causes initial value dependence [17, Chapter 3], the problem of finding an appropriate setting of the low-rank structure is highly nontrivial

37th Conference on Neural Information Processing Systems (NeurIPS 2023).

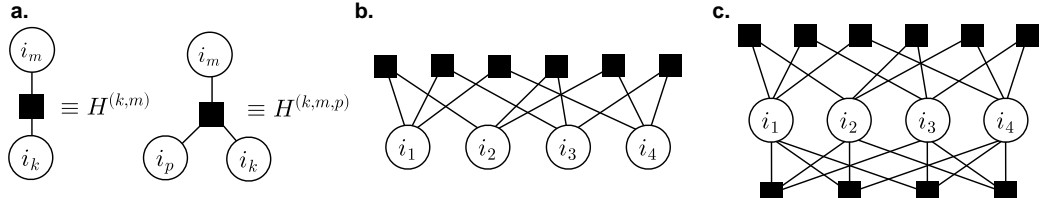

Figure 1: Interaction representations corresponding to (**a**) second- and third-order energy (**b**) two- and (**c**) three-body approximation.

in practice as it is hard to locate the cause if the decomposition does not perform well. As a result, in order to find proper structure and rank, the user must often perform decomposition multiple times with various settings, which is time- and memory-consuming. Although there is an attempt to avoid rank tuning by approximating it by trace norms [2], it also requires another hyperparameter: the weights of the unfolding tensor. Hence, such an approach does not fundamentally solve the problem.

Instead of the low-rank structure that has been the focus of attention in the past, in the present paper we propose a novel formulation of non-negative tensor decomposition, called *many-body approximation*, that focuses on the relationship among modes of tensors. The structure of decomposition can be naturally determined based on the existence of the interactions between modes. The proposed method requires only the decomposition structure and does not require the rank value, which traditional decomposition methods also require as a hyperparameter and often suffer to determine.

To describe interactions between modes, we follow the standard strategy in statistical mechanics that uses an energy function $\mathcal{H}(\cdot)$ to treat interactions and considers the corresponding distribution $\exp\left(-\mathcal{H}(\cdot)\right)$. This model is known to be an energy-based model in machine learning [19] and is exploited in tensor decomposition as Legendre decomposition [36]. Technically, it parameterizes a tensor as a discrete probability distribution and reduces the number of parameters by enforcing some of them to be zero in optimization. We explore this energy-based approach further and discover the family of parameter sets that represent interactions between modes in the energy function $\mathcal{H}(\cdot)$. How to choose non-zero parameters in Legendre decomposition has been an open problem; we start by addressing this problem and propose many-body approximation as a special case of Legendre decomposition. Moreover, although Legendre decomposition is not factorization of tensors in general, our proposal always offers factorization, which can reveal patterns in tensors. Since the advantage of Legendre decomposition is inherited to our proposal, many-body approximation can be achieved by convex optimization that globally minimizes the Kullback–Leibler (KL) divergence [18].

Furthermore, we introduce a way of representing mode interactions, that visualizes the presence or absence of interactions between modes as a diagram. We discuss the relation to the tensor network and point out that an operation called coarse-grained transformation [20] – in which multiple tensors are viewed as a new tensor – reveals unexpected relationship between the proposed method and existing methods such as tensor ring and tensor tree decomposition.

We summarize our contribution as follows:
- By focusing on the interaction between modes of tensors, we introduce rank-free tensor decomposition, called many-body approximation. This decomposition is achieved by convex optimization.
- We present a way of describing tensor many-body approximation called interaction representation, which is a diagram that shows interactions within a tensor and can be transformed into tensor networks.
- Many-body approximation can perform tensor completion via the $em$-algorithm, which empirically shows better performance than low-rank based existing methods.

## 2 Many-body Approximation for Tensors

Our proposal, many-body approximation, is based on the formulation of Legendre decomposition for tensors, which we first review in Section 2.1. Then we introduce interactions between modes and its visual representation in Section 2.2, many-body approximation in Section 2.3, and transformation of interaction representation into tensor networks in Section 2.4. In the following discussion, we consider $D$-order non-negative tensors with the size $(I_1, \ldots, I_D)$, and we denote by $[K] = \{1, 2, \ldots, K\}$ for a positive integer $K$. We assume the sum of all elements in a given tensor to be 1 for simplicity,

while this assumption can be eliminated using the general property of the Kullback–Leibler (KL) divergence, $\lambda D_{KL}(\mathcal{P}, \mathcal{Q}) = D_{KL}(\lambda \mathcal{P}, \lambda \mathcal{Q})$, for any $\lambda \in \mathbb{R}_{>0}$.

## 2.1 Reminder to Legendre Decomposition and its optimization

Legendre decomposition is a method for decomposing a non-negative tensor by regarding the tensor as a discrete distribution and representing it with a limited number of parameters. We describe a non-negative tensor $\mathcal{P}$ using parameters $\boldsymbol{\theta} = (\theta_{2,\ldots,1}, \ldots, \theta_{I_1,\ldots,I_D})$ and its energy function $\mathcal{H}$ as

$$\mathcal{P}_{i_1,\ldots,i_D} = \exp\left(-\mathcal{H}_{i_1,\ldots,i_D}\right), \quad \mathcal{H}_{i_1,\ldots,i_D} = -\sum_{i'_1=1}^{i_1} \cdots \sum_{i'_D=1}^{i_D} \theta_{i'_1,\ldots,i'_D}, \tag{1}$$

where $\theta_{1,\ldots,1}$ has a role of normalization. It is clear that the index set of a tensor corresponds to sample space of a distribution and the value of each entry of the tensor is regarded as the probability of realizing the corresponding index [35].

As we see in Equation (1), we can uniquely identify tensors from the parameters $\boldsymbol{\theta}$. Inversely, we can compute $\boldsymbol{\theta}$ from a given tensor as

$$\theta_{i_1,\ldots,i_D} = \sum_{i'_1=1}^{I_1} \cdots \sum_{i'_D=1}^{I_D} \mu_{i_1,\ldots,i_D}^{i'_1,\ldots,i'_D} \log \mathcal{P}_{i'_1,\ldots,i'_D} \tag{2}$$

using the Möbius function $\mu : S \times S \to \{-1, 0, +1\}$, where $S$ is the set of indices, defined inductively as follows:

$$\mu_{i_1,\ldots,i_D}^{i'_1,\ldots,i'_D} = \begin{cases} 1 & \text{if } i_d = i'_d, \forall d \in [D], \\ -\prod_{d=1}^{D} \sum_{j_d=i_d}^{i'_d-1} \mu_{i_1,\ldots,i_D}^{j_1,\ldots,j_D} & \text{else if } i_d \leq i'_d, \forall d \in [D], \\ 0 & \text{otherwise.} \end{cases}$$

Since distribution described by Equation (1) belongs to the exponential family, $\boldsymbol{\theta}$ corresponds to natural parameters of the exponential family, and we can also identify each tensor by expectation parameters $\boldsymbol{\eta} = (\eta_{2,\ldots,1}, \ldots, \eta_{I_1,\ldots,I_D})$ using Möbius inversion as

$$\eta_{i_1,\ldots,i_D} = \sum_{i'_1=i_1}^{I_1} \cdots \sum_{i'_D=i_D}^{I_D} \mathcal{P}_{i'_1,\ldots,i'_D}, \quad \mathcal{P}_{i_1,\ldots,i_D} = \sum_{i'_1=1}^{I_1} \cdots \sum_{i'_D=1}^{I_D} \mu_{i_1,\ldots,i_D}^{i'_1,\ldots,i'_D} \eta_{i'_1,\ldots,i'_D}, \tag{3}$$

where $\eta_{1,\ldots,1} = 1$ because of normalization. See Section A in Supplement for examples of the above calculation. Since distribution is determined by specifying either $\theta$-parameters or $\eta$-parameters, they form two coordinate systems, called the $\theta$-coordinate system and the $\eta$-coordinate system, respectively.

### 2.1.1 Optimization

Legendre decomposition approximates a tensor by setting some $\theta$ values to be zero, which corresponds to dropping some parameters for regularization. It achieves convex optimization using the dual flatness of $\theta$- and $\eta$-coordinate systems. Let $B$ be the set of indices of $\theta$ parameters that are not imposed to be 0. Then Legendre decomposition coincides with a projection of a given nonnegative tensor $\mathcal{P}$ onto the subspace $\mathcal{B} = \{\mathcal{Q} \mid \theta_{i_1,\ldots,i_D} = 0 \text{ if } (i_1, \ldots, i_D) \notin B \text{ for } \theta\text{-parameters of } \mathcal{Q}\}$.

Let us consider projection of a given tensor $\mathcal{P}$ onto $\mathcal{B}$. The space of probability distributions, which is not a Euclidean space, is studied in information geometry. By taking geometry of probability distributions into account, we can introduce the concept of flatness for a set of tensors. The subspace $\mathcal{B}$ is $e$-flat if the logarithmic combination, or called $e$-geodesic, $\mathcal{R} \in \{(1-t) \log \mathcal{Q}_1 + t \log \mathcal{Q}_2 - \phi(t) \mid 0 < t < 1\}$ of any two points $\mathcal{Q}_1, \mathcal{Q}_2 \in \mathcal{B}$ is included in the subspace $\mathcal{B}$, where $\phi(t)$ is a normalizer. There is always a unique point $\overline{\mathcal{P}}$ on the $e$-flat subspace that minimizes the KL divergence from any point $\mathcal{P}$.

$$\overline{\mathcal{P}} = \underset{\mathcal{Q}; \mathcal{Q} \in \mathcal{B}}{\arg\min} \, D_{KL}(\mathcal{P}, \mathcal{Q}). \tag{4}$$

This projection is called the $m$-projection. The $m$-projection onto an $e$-flat subspace is convex optimization. We define two vectors $\boldsymbol{\theta}^B = (\theta_b)_{b \in B}$ and $\boldsymbol{\eta}^B = (\eta_b)_{b \in B}$ and write the dimensionality of these vectors as $|B|$ since it is equal to the cardinality of $B$. The derivative of the KL divergence and the Hessian matrix $G \in \mathbb{R}^{|B| \times |B|}$ are given as

$$\frac{\partial}{\partial \boldsymbol{\theta}^B} D_{KL}(\mathcal{P}, \mathcal{Q}) = \boldsymbol{\eta}^B - \hat{\boldsymbol{\eta}}^B, \quad \mathbf{G}_{u,v} = \eta_{\max(i_1,j_1),\ldots,\max(i_D,j_D)} - \eta_{i_1,\ldots,i_D} \eta_{j_1,\ldots,j_D}, \quad (5)$$

where $\boldsymbol{\eta}^B$ and $\hat{\boldsymbol{\eta}}^B$ are the expectation parameters of $\mathcal{Q}$ and $\mathcal{P}$, respectively, and $u = (i_1,\ldots,i_D), v = (j_1,\ldots,j_D) \in B$. This matrix $\mathbf{G}$ is also known as the negative Fisher information matrix. Using gradient descent with second-order derivative, we can update $\boldsymbol{\theta}^B$ in each iteration $t$ as

$$\boldsymbol{\theta}_{t+1}^B = \boldsymbol{\theta}_t^B - \mathbf{G}^{-1}(\boldsymbol{\eta}_t^B - \hat{\boldsymbol{\eta}}^B). \quad (6)$$

The distribution $\mathcal{Q}_{t+1}$ is calculated from the updated natural parameters $\boldsymbol{\theta}_{t+1}$. This step finds a point $\mathcal{Q}_{t+1} \in \boldsymbol{\mathcal{B}}$ that is closer to the destination $\overline{\mathcal{P}}$ along with the $e$-geodesic from $\mathcal{Q}_t$ to $\overline{\mathcal{P}}$. We can also calculate the expected value parameters $\boldsymbol{\eta}_{t+1}$ from the distribution. By repeating this process until convergence, we can always find the globally optimal solution satisfying Equation (4). See Section A in Supplement for more detail on the optimization. The problem of how to design and interpret $B$ has not been explored before, and we firstly give the reasoning of $B$ as the presence or absence of particular interactions between modes, which naturally leads to the design guideline of $B$. The computational complexity of Legendre decomposition is $\mathcal{O}(\gamma |B|^3)$, where $\gamma$ is the number of iterations.

## 2.2  Interaction and its representation of tensors

In this subsection, we introduce interactions between modes and its visual representation to prepare for many-body approximation. The following discussion enables us to intuitively describe relationships between modes and formulate our novel rank-free tensor decomposition.

First, we introduce $n$-body parameters, a generalized concept of one-body and two-body parameters in [11]. In any $n$-body parameter, there are $n$ non-one indices; for example, $\theta_{1,2,1,1}$ is a one-body parameter, $\theta_{4,3,1,1}$ is a two-body parameter, and $\theta_{1,2,4,3}$ is a three-body parameter. We also use the following notation for $n$-body parameters:

$$\theta_{i_k}^{(k)} = \theta_{1,\ldots,1,i_k,1,\ldots,1}, \quad \theta_{i_k,i_m}^{(k,m)} = \theta_{1,\ldots,1,i_k,1,\ldots,1,i_m,1,\ldots,1}, \quad \theta_{i_k,i_m,i_p}^{(k,m,p)} = \theta_{1,\ldots,i_k,\ldots,i_m,\ldots,i_p,\ldots,1},$$

for $n = 1, 2$ and $3$, respectively. We write the energy function $\mathcal{H}$ with $n$-body parameters as

$$\mathcal{H}_{i_1,\cdots,i_D} = H_0 + \sum_{m=1}^{D} H_{i_m}^{(m)} + \sum_{m=1}^{D} \sum_{k=1}^{m-1} H_{i_k,i_m}^{(k,m)} + \sum_{m=1}^{D} \sum_{k=1}^{m-1} \sum_{p=1}^{k-1} H_{i_p,i_k,i_m}^{(p,k,m)} + \cdots + H_{i_1,\ldots,i_D}^{(1,\ldots,D)}, \quad (7)$$

where the $n$-th order energy is introduced as

$$H_{i_{l_1},\ldots,i_{l_n}}^{(l_1,\ldots,l_n)} = -\sum_{i'_{l_1}=2}^{i_{l_1}} \cdots \sum_{i'_{l_n}=2}^{i_{l_n}} \theta_{i'_{l_1},\ldots,i'_{l_n}}^{(l_1,\ldots,l_n)}. \quad (8)$$

For simplicity, we suppose that $1 \le l_1 < l_2 < \cdots < l_n \le D$ holds. We set $H_0 = -\theta_{1,\ldots,1}$. We say that an $n$-body interaction exists between modes $l_1, \ldots, l_n$ if there are indices $i_{l_1}, \ldots, i_{l_n}$ satisfying $H_{i_{l_1},\ldots,i_{l_n}}^{(l_1,\ldots,l_n)} \ne 0$. The first term, $H_0$ in Equation (7), is called the normalized factor or the partition function. The terms $H^{(k)}$ are called bias in machine learning and magnetic field or self-energy in statistical physics. The terms $H^{(k,m)}$ are called the weight of the Boltzmann machine in machine learning and two-body interaction or electron-electron interaction in physics.

To visualize the existence of interactions within a tensor, we introduce a diagram called *interaction representation*, which is inspired by factor graphs in graphical modeling [3, Chapter 8]. The graphical representation of the product of tensors is widely known as tensor networks. However, displaying the relations between modes of a tensor as a factor graph is our novel approach. We represent the $n$-body interaction as a black square, ■, connected with $n$ modes. We describe examples of the

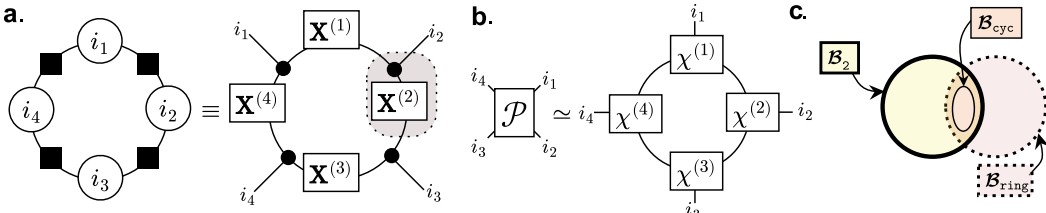

Figure 2: (**a**)(left) Interaction representation of an example of cyclic two-body approximation and (**a**)(right) its transformed tensor network for $D = 4$. Each tensor is enclosed by a square and each mode is enclosed by a circle. A black circle ● is a hyper diagonal tensor. Edges through ■ between modes mean interaction existence. (**b**) Tensor network of tensor ring decomposition. (**c**) The relationship among solution spaces of tensor-ring decomposition $\mathcal{B}_{\mathrm{ring}}$, two-body approximation $\mathcal{B}_2$, and cyclic two-body approximation $\mathcal{B}_{\mathrm{cyc}}$.

two-body interaction between modes $(k, m)$ and the three-body interaction among modes $(k, m, p)$ in Figure 1(**a**). Combining these interactions, the diagram for the energy function including all two-body interactions is shown in Figure 1(**b**), and all two-body and three-body interactions are included in Figure 1(**c**) for $D = 4$. This visualization allows us to intuitively understand the relationship between modes of tensors. For simplicity, we abbreviate one-body interactions in the diagrams, while we always assume them. Once interaction representation is given, we can determine the corresponding decomposition of tensors.

In Boltzmann machines, we usually consider binary (two-level) variables and their second-order interactions. In our proposal, we consider multi-level $D$ variables, each of which can take a natural number from 1 to $I_d$ for $d \in [D]$. Moreover, higher-order interactions among them are allowed. Therefore, our proposal is a form of a multi-level extension of Boltzmann machines with higher-order interaction, where each node of Boltzmann machines corresponds to the tensor mode. In the following section, we reduce some of $n$-body interactions, – that is, $H_{i_{l_1},\ldots,i_{l_n}}^{(l_1,\ldots,l_n)} = 0$ – by fixing each parameter $\theta_{i_{l_1},\ldots,i_{l_n}}^{(l_1,\ldots,l_n)} = 0$ for all indices $(i_{l_1}, \ldots, i_{l_n}) \in \{2, \ldots, I_{l_1}\} \times \cdots \times \{2, \ldots, I_{l_n}\}$.

### 2.3 Many-body approximation

We approximate a given tensor by assuming the existence of dominant interactions between the modes of the tensor and ignoring other interactions. Since this operation can be understood as enforcing some natural parameters of the distribution to be zero, it can be achieved by convex optimization through the theory of Legendre decomposition. We discuss how we can choose dominant interactions in Section 3.1.

As an example, we consider two types of approximations of a nonnegative tensor $\mathcal{P}$ by tensors represented in Figure 1(**b, c**). If all energies greater than second-order or those greater than third-order in Equation (7) are ignored – that is, $H_{i_{l_1},\ldots,i_{l_n}}^{(l_1,\ldots,l_n)} = 0$ for $n > 2$ or $n > 3$ – the tensor $\mathcal{P}_{i_1,i_2,i_3,i_4}$ is approximated as follows:

$$\mathcal{P}_{i_1,i_2,i_3,i_4} \simeq \mathcal{P}_{i_1,i_2,i_3,i_4}^{\leq 2} = \mathbf{X}_{i_1,i_2}^{(1,2)}\mathbf{X}_{i_1,i_3}^{(1,3)}\mathbf{X}_{i_1,i_4}^{(1,4)}\mathbf{X}_{i_2,i_3}^{(2,3)}\mathbf{X}_{i_2,i_4}^{(2,4)}\mathbf{X}_{i_3,i_4}^{(3,4)},$$

$$\mathcal{P}_{i_1,i_2,i_3,i_4} \simeq \mathcal{P}_{i_1,i_2,i_3,i_4}^{\leq 3} = \chi_{i_1,i_2,i_3}^{(1,2,3)}\chi_{i_1,i_2,i_4}^{(1,2,4)}\chi_{i_1,i_3,i_4}^{(1,3,4)}\chi_{i_2,i_3,i_4}^{(2,3,4)},$$

where each of matrices and tensors on the right-hand side is represented as

$$\mathbf{X}_{i_k,i_m}^{(k,m)} = \frac{1}{\sqrt[6]{Z}}\exp\left(\frac{1}{3}H_{i_k}^{(k)} + H_{i_k,i_m}^{(k,m)} + \frac{1}{3}H_{i_m}^{(m)}\right),$$

$$\chi_{i_k,i_m,i_p}^{(k,m,p)} = \frac{1}{\sqrt[4]{Z}}\exp\left(\frac{H_{i_k}^{(k)}}{3} + \frac{H_{i_m}^{(m)}}{3} + \frac{H_{i_p}^{(p)}}{3} + \frac{1}{2}H_{i_k,i_m}^{(k,m)} + \frac{1}{2}H_{i_m,i_p}^{(m,p)} + \frac{1}{2}H_{i_k,i_p}^{(k,p)} + H_{i_k,i_m,i_p}^{(k,m,p)}\right).$$

The partition function, or the normalization factor, is given as $Z = \exp\left(-\theta_{1,1,1,1}\right)$, which does not depend on indices $(i_1, i_2, i_3, i_4)$. Each $\mathbf{X}^{(k,m)}$ (resp. $\chi^{(k,m,p)}$) is a factorized representation for the relationship between $k$-th and $m$-th (resp. $k$-th, $m$-th and $p$-th) modes. Although our model can be

transformed into a linear model by taking the logarithm, our convex formulation enables us to find the optimal solution more stable than traditional linear low-rank-based nonconvex approaches. Since we do not impose any low-rankness, factorized representations, such as $\mathbf{X}^{(k,m)}$ and $\chi^{(k,m,p)}$, can be full-rank matrices or tensors.

For a given tensor $\mathcal{P}$, we define its $m$-*body approximation* $\mathcal{P}^{\leq m}$ as the optimal solution of Equation (4) – that is, $\mathcal{P}^{\leq m} = \arg\min_{\mathcal{Q} \in \mathcal{B}_m} D_{KL}(\mathcal{P}, \mathcal{Q})$ – where the solution space $\mathcal{B}_m$ is given as

$$\mathcal{B}_m = \left\{ \mathcal{Q} \mid \theta_{i_1,\ldots,i_D} = 0 \text{ if } \theta_{i_1,\ldots,i_D} \text{ is } n(> m)\text{-body parameters for } \theta\text{-parameters of } \mathcal{Q} \right\}.$$

Note that $\mathcal{P}^{\leq D} = \mathcal{P}$ always holds. For any natural number $m < D$, it holds that $\mathcal{B}_m \subseteq \mathcal{B}_{m+1}$. Interestingly, the two-body approximation for a non-negative tensor with $I_1 = \cdots = I_D = 2$ is equivalent to approximating the empirical distribution with the fully connected Boltzmann machine.

In the above discussion, we consider many-body approximation with all the $n$-body parameters, while we can relax this constraint and allow the use of only a part of $n$-body parameters in approximation. Let us consider the situation where only one-body interaction and two-body interaction between modes $(d, d+1)$ exist for all $d \in [D]$ ($D+1$ implies $1$ for simplicity). Figure 2(**a**) shows the interaction representation of the approximated tensor. As we can confirm by substituting $0$ for $H_{i_k,i_l}^{(k,l)}$ if $l \neq k+1$, we can describe the approximated tensor as

$$\mathcal{P}_{i_1,\ldots,i_D} \simeq \mathcal{P}_{i_1,\ldots,i_D}^{\mathrm{cyc}} = \mathbf{X}_{i_1,i_2}^{(1)} \mathbf{X}_{i_2,i_3}^{(2)} \ldots \mathbf{X}_{i_D,i_1}^{(D)}, \tag{9}$$

where

$$\mathbf{X}_{i_k,i_{k+1}}^{(k)} = \frac{1}{\sqrt[D]{Z}} \exp\left( \frac{1}{2} H_{i_k}^{(k)} + H_{i_k,i_{k+1}}^{(k,k+1)} + \frac{1}{2} H_{i_{k+1}}^{(k+1)} \right)$$

with the normalization factor $Z = \exp\left(-\theta_{1,\ldots,1}\right)$. When the tensor $\mathcal{P}$ is approximated by $\mathcal{P}^{\mathrm{cyc}}$, the set $B$ contains only all one-body parameters and two-body parameters $\theta_{i_d,i_{d+1}}^{(d,d+1)}$ for $d \in [D]$. We call this approximation *cyclic two-body approximation* since the order of indices in Equation (9) is cyclic. It holds that $\mathcal{B}_{\mathrm{cyc}} \subseteq \mathcal{B}_2$ for the solution space of cyclic two-body approximation $\mathcal{B}_{\mathrm{cyc}}$ (Figure 2(**c**)).

## 2.4 Connection to tensor networks

Our tensor interaction representation is an undirected graph that focuses on the relationship between modes. By contrast, tensor networks, which are well known as diagrams that focus on smaller tensors after decomposition, represent a tensor as an undirected graph, whose nodes correspond to matrices or tensors and edges to summation over a mode in tensor products [6].

We provide examples in our representation that can be converted to tensor networks, which implies a tight connection between our representation and tensor networks. For the conversion, we use a hyper-diagonal tensor $\mathcal{I}$ defined as $\mathcal{I}_{ijk} = \delta_{ij}\delta_{jk}\delta_{ki}$, where $\delta_{ij} = 1$ if $i = j$ and $0$ otherwise. The tensor $\mathcal{I}$ is often represented by $\bullet$ in tensor networks. In the community of tensor networks, the tensor $\mathcal{I}$ appears in the CNOT gate and a special case of the Z spider [28]. The tensor network in Figure 2(**a**) represents Equation (9) for $D = 4$.

A remarkable finding is that the converted tensor network representation of cyclic two-body approximation and the tensor network of tensor ring decomposition, whose tensor network is shown in Figure 2(**b**), have a similar structure, despite their different modeling. If we consider the region enclosed by the dotted line in Figure 2(**a**) as a new tensor, the tensor network of the cyclic two-body approximation coincides with the tensor network of the tensor ring decomposition shown in Figure 2(**b**). This operation, in which multiple tensors are regarded as a new tensor in a tensor network, is called coarse-graining transformation [9].

Formally, cyclic two-body approximation coincides with tensor ring decomposition with a specific constraint as described below. Non-negative tensor ring decomposition approximates a given tensor $\mathcal{P} \in \mathbb{R}_{\geq 0}^{I_1 \times \cdots \times I_D}$ with $D$ core tensors $\chi^{(1)}, \ldots, \chi^{(D)} \in \mathbb{R}_{\geq 0}^{R_{d-1} \times I_d \times R_d}$ as

$$\mathcal{P}_{i_1,\ldots,i_D} \simeq \sum_{r_1=1}^{R_1} \cdots \sum_{r_D=1}^{R_D} \chi_{r_D,i_1,r_1}^{(1)} \cdots \chi_{r_{D-1},i_D,r_D}^{(D)}, \tag{10}$$

where $(R_1, \ldots, R_D)$ is called the tensor ring rank. The cyclic two-body approximation also approximates the tensor $\mathcal{P}$ in the form of Equation (10), imposing an additional constraint that each core tensor $\chi^{(d)}$ is decomposed as

$$\chi^{(d)}_{r_{d-1}, i_d, r_d} = \sum_{m_d=1}^{I_d} \mathbf{X}^{(d)}_{r_{d-1}, m_d} \mathcal{I}_{m_d, i_d, r_d} \tag{11}$$

for each $d \in [D]$. We assume $r_0 = r_D$ for simplicity. We obtain Equation (9) by substituting Equation (11) into Equation (10).

This constraint enables us to perform convex optimization. This means that we find a subclass $\mathcal{B}_{\mathrm{cyc}}$ that can be solved by convex optimization in tensor ring decomposition, which has suffered from the difficulty of non-convex optimization. In addition, this is simultaneously a subclass of two-body approximation, as shown in Figure 2(**c**).

From Kronecker's delta $\delta$, $r_d = i_d$ holds in Equation (11); thus, $\chi^{(d)}$ is a tensor with the size $(I_{d-1}, I_d, I_d)$. Tensor ring rank after the cyclic two-body approximation is $(I_1, \ldots, I_D)$ since the size of core tensors coincides with tensor ring rank. This result firstly reveals the relationship between Legendre decomposition and low-rank approximation via tensor networks.

Here we compare the number of parameters of cyclic two-body approximation and that of tensor ring decomposition. The number of elements of an input tensor is $I_1 I_2 \cdots I_D$. After cyclic two-body approximation, the number $|B|$ of parameters is given as

$$|B| = 1 + \sum_{d=1}^{D}(I_d - 1) + \sum_{d=1}^{D}(I_d - 1)(I_{d+1} - 1), \tag{12}$$

where we assume $I_{D+1} = I_1$. The first term is for the normalizer, the second for the number of one-body parameters, and the final term for the number of two-body parameters. In contrast, in the tensor ring decomposition with the target rank $(R_1, \ldots, R_D)$, the number of parameters is given as $\sum_{d=1}^{D} R_d I_d R_{d+1}$. The ratio of the number of parameters of these two methods is proportional to $I/R^2$ if we assume $R_d = R$ and $I_d = I$ for all $d \in [D]$ for simplicity. Therefore, when the target rank is small and the size of the input tensor is large, the proposed method has more parameters than the tensor ring decomposition. The correspondence of many-body approximation to existing low-rank approximation is not limited to tensor ring decomposition. We also provide another example of the correspondence for tensor tree decomposition in Section C.2 in Supplement.

### 2.5 Many-body approximation as generalization of mean-field approximation

Any tensor $\mathcal{P}$ can be represented by vectors $\boldsymbol{x}^{(1)}, \ldots, \boldsymbol{x}^{(D)} \in \mathbb{R}^{I_d}$ as $\mathcal{P}_{i_1, \ldots, i_D} = x_{i_1}^{(1)} x_{i_2}^{(2)} \ldots x_{i_D}^{(D)}$ if and only if all $n(\geq 2)$-body $\theta$-parameters are 0 [10, 12]. The right-hand side is equal to the Kronecker product of $D$ vectors $\boldsymbol{x}^{(1)}, \ldots, \boldsymbol{x}^{(D)}$; therefore, this approximation is equivalent to the rank-1 approximation since the rank of the tensor that can be represented by the Kronecker product is always 1, which is also known to correspond to mean-field approximation. In this study, we propose many-body approximation by relaxing the condition for the mean-field approximation that ignores $n(\geq 2)$-body interactions. Therefore many-body approximation is generalization of rank-1 approximation and mean-field approximation.

The discussion above argues that the one-body approximation – that is, the rank-1 approximation that minimizes the KL divergence from a given tensor – is a convex problem. Although finding the rank-1 tensor that minimizes the Frobenius norm from a given tensor is known to be an NP-hard problem [14], it has been reported that finding the rank-1 tensor that minimizes the KL divergence from a given tensor is a convex problem [16]. Therefore, our claim is not inconsistent with those existing studies. It should be also noted that, except for the rank-1 case, there is no guarantee that traditional low-rank decomposition that optimizes the KL divergence is a convex problem [5, 13].

### 2.6 Low-body tensor completion

Since our proposal is an alternative to the existing low-rank approximation, we can replace it with our many-body approximation in practical applications. As a representative application, we demonstrate

**Algorithm 1:** Low-body tensor completion

---

LBTC($\mathcal{T}$, $B$, $\Omega$)                                                   // $\Omega$ is observed indices
  │ $t \leftarrow 1$
  │ Initialize $\mathcal{P}^t$                                                       // e.g. a random tensor
  │ $\mathcal{P}^t_\Omega \leftarrow \mathcal{T}_\Omega$
  │ **repeat**
  │   │ $\mathcal{P}^{t+1} \leftarrow$ MANYBODYAPPROXIMATION $(\mathcal{P}^t, B)$      // $m$-step, See Algorithm 2
  │   │ $\mathcal{P}^{t+1}_\Omega \leftarrow \mathcal{T}_\Omega$                      // $e$-step
  │   │ $\mathrm{res}^t \leftarrow \|\mathcal{P}^{t+1} - \mathcal{P}^t\|_F / \|\mathcal{P}^t\|_F$
  │   │ $t \leftarrow t + 1$
  │ **until** $\mathrm{res}^t - \mathrm{res}^{t-1} < \epsilon$ *and* $t > 2$          // We set $\epsilon = 10^{-5}$ in our implementation;
  └ **return** $\mathcal{P}$

---

Table 1: Recovery fit score for tensor completion.

| | Scenario 1 | Scenario 2 | Scenario 3 |
|---|---|---|---|
| LBTC | **0.948**±0.00152 | **0.917**±0.000577 | **0.874**±0.00153 |
| HaLRTC | 0.864 | 0.842 | 0.825 |
| SiLRTC | 0.863 | 0.841 | 0.820 |
| SiLRTCTT | 0.948 | 0.915 | 0.868 |
| PTRCRW | 0.945±0.0000 | 0.901±0.0000 | 0.844±0.0000 |

missing value estimation by the $em$-algorithm [26], where we replace the $m$-step of low-rank approximation with many-body approximation. We call this approach low-body tensor completion (LBTC). The $em$-algorithm with tensor low-rank approximation does not guarantee the uniqueness of the $m$-step because the model space is not flat, while LBTC ensures that the model space is flat, and thus the $m$-step of LBTC is unique. See more discussion of information geometry for LBTC in Section C.4. We provide its pseudo-code in Algorithm 1. Instead of hyper-parameters related to ranks that traditional tensor completion methods require, LBTC requires only information about interactions to be used, but it can be intuitively tuned, as also seen in Section 3.1. We examine the performance of LBTC in Section 3.2.

## 3 Experiments

We conducted three experiments to see the usefulness and effectiveness of many-body approximation. Datasets and implementation details are available in Supplement.

### 3.1 An example of tuning interaction

We extracted 10 color images from Columbia Object Image Library (COIL-100) [27] and constructed a tensor $\mathcal{P} \in \mathbb{R}^{40 \times 40 \times 3 \times 10}_{\geq 0}$, where each mode represents the width, height, color, or image index, respectively. Then we decomposed $\mathcal{P}$ with varying chosen interactions and showed the reconstructed images. Figure 3(**a**) shows $\mathcal{P}^{\leq 4}$, which coincides with the input tensor $\mathcal{P}$ itself since its order is four. Reconstruction quality gradually dropped when we reduced interactions from Figure 3(**a**) to Figure 3(**d**). We can explicitly control the trend of reconstruction by manually choosing interaction, which is one of characteristics of the proposal. In Figure 3(**c**), each reconstructed image has a monotone taste because the color is not directly interacted with the pixel position (width and height). In Figure 3(**d**), although the whole tensor is monotone because the color is independent of any other modes, the reconstruction keeps the objects' shapes since the interaction among width, height, and image index are retained. See Figure 7 in Supplement for the role of the interaction. Interactions between modes are interpretable, in contrast to the rank of tensors, which is often difficult to interpret. Therefore, tuning of interactions is more intuitive than that of ranks.

We also visualize the interaction between color and image as a heatmap of a matrix $\mathbf{X}^{(c,i)} \in \mathbb{R}^{3 \times 10}$ in Figure 3(**e**), which is obtained in the case of Figure 3(**c**). We normalize each column of $\mathbf{X}^{(c,i)}$ with the

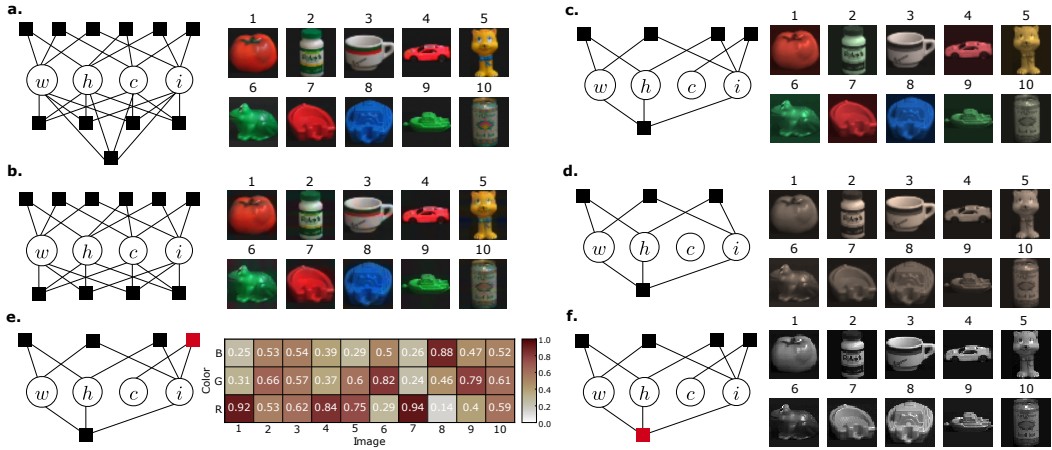

Figure 3: (**a,b**) Four-body and three-body approximation for the COIL dataset. (**c,d**) Appropriate selection of interactions controls the richness of color. Visualization of interaction between (**e**) color and image index, and (**f**) width, height, and image index.

L2 norm for better visualization. As expected, indices corresponding to red objects (that is, 1, 4, and 7) have large values in the third red-color row. We also visualize the interaction among width, height, and image index, which corresponds to a tensor $\chi^{(w,h,i)} \in \mathbb{R}^{40 \times 40 \times 10}$ without any information of colors, as a series of heatmaps in Figure 3(**f**). They provide the gray-scale reconstruction of images. The above discussion implies that many-body approximation captures patterns or features of inputs.

## 3.2 Tensor completion by many-body approximation

To examine the performance of LBTC, we conducted experiments on a real-world dataset: the traffic speed dataset in District 7, Los Angeles County, collected from PeMS [31]. This data was obtained from fixed-point observations of vehicle speeds every five minutes for 28 days. We made $28 \times 24 \times 12 \times 4$ tensor $\mathcal{T}$, whose modes correspond to date, hour, minute, and lane. As practical situations, we assumeed that a disorder of a lane's speedometer causes random and continuous deficiencies. We prepared three patterns of missing cases: a case with a disorder in the speedometer in Lane 1 (Scenario 1), a case with disorders in the speedometers in Lanes 1, 2, and 3 (Scenario 2), and a case with disorders in all speedometers (Scenario 3). The missing rates for scenarios 1, 2, and 3 were 9 percent, 27 percent, and 34 percent, respectively. We evaluate the recovery fit score $1 - \|\mathcal{T}_\tau - \overline{\mathcal{T}}_\tau\|_F / \|\mathcal{T}_\tau\|_F$ for the reconstruction $\overline{\mathcal{T}}$ and missing indices $\tau$.

In the proposal, we need to choose interactions to use. Intuitively, the interaction between date and minute and that between minute and lane seem irrelevant. Therefore, we used all the two-body interactions except for the above two interactions. In addition, the interaction among date, hour, and minute is also included because it is intuitively relevant.

As baseline methods, we use SiLRTC, HaLRTC [21], SiLRTC-TT [2], and PTRCRW [38]. SiLRTC-TT and PTRCRW are based on tensor train and ring decompositions, respectively, and PTRCRW is known as the state of the art. The reconstructions are available in Figure 8 in Supplement with ground truth. We tuned hyperparameters of baseline methods and compared their best results with LBTC. The resulting scores (see Table 1) show that our method is superior to other methods. Moreover, we can intuitively tune interactions in the proposal, while other baselines need typical hyperparameter tuning as seen in Figure 14 in Supplement. In our method, interactions among non-associated modes could worsen the performance as seen in Table 2 in Supplement. Since the proposal and PTRCRW are non-convex optimizations, we ran each of them 10 times with random initialization and reported mean values and the standard error. Many-body approximation itself is always convex, while our tensor completion algorithm LBTC is non-convex due to the use of the *em*-algorithm.

### 3.3 Comparison with ring decomposition

As seen in Section 2.4, many-body approximation has a close connection to low-rank approximation. For example, in tensor ring decomposition, if we impose the constraint that decomposed smaller tensors can be represented as products with hyper-diagonal tensors $\mathcal{I}$, this decomposition is equivalent to a cyclic two-body approximation. Therefore, to examine our conjecture that cyclic two-body approximation is as capable of approximating as tensor ring decomposition, we empirically examined the effectiveness of cyclic two-body approximation compared with tensor ring decomposition. As baselines, we used five existing methods of non-negative tensor ring decomposition: NTR-APG, NTR-HALS, NTR-MU, NTR-MM, and NTR-lraMM [39, 40]. We also describe the efficiency in Supplement.

We evaluated the approximation performance by the relative error $\|\mathcal{T} - \overline{\mathcal{T}}\|_F/\|\mathcal{T}\|_F$ for an input tensor $\mathcal{T}$ and a reconstructed tensor $\overline{\mathcal{T}}$. Since all the existing methods are based on nonconvex optimization, we plotted the best score (minimum relative error) among five restarts with random initialization. In contrast, the score of our method is obtained by a single run as it is convex optimization and such restarts are fundamentally unnecessary.

**Synthetic data** We performed experiments on two synthetic datasets with low tensor ring rank. We create sixth- and seventhth-order tensors with ring ranks $(15, ..., 15)$ and $(10, ..., 10)$, respectively. In Figure 4(**a**), relative errors are plotted with gradually increasing the target rank of the tensor ring decomposition, which is compared to the score of our method, plotted as the cross point of horizontal and vertical red dotted lines. Please note that our method does not have the concept of the rank, so the score of our method is invariant to changes of the target rank unlike other methods. If the cross point of red dotted lines is lower than other lines, the proposed method is better than other methods. In both experiments, the proposed method is superior to comparison partners in terms of the reconstruction error.

**Real data** Next, we evaluated our method on real data. `TT_ChartRes` and `TT_Origami` are seventh-order tensors that are produced from TokyoTech Hyperspectral Image Dataset [23, 24]. Each tensor has been reshaped to reduce the computational complexity. As seen in Fig-

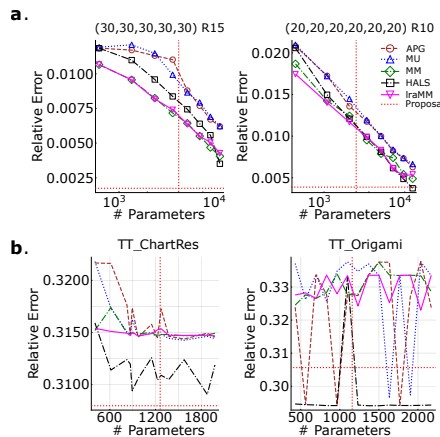

Figure 4: Experimental results for (**a**) synthetic data and (**b**) real datasets. The vertical red dotted line is $|B|$ (see Equation (12)).

ure 4(**b**), the proposed method keeps the competitive relative errors. In baselines, a slight change of the target rank can induce a significant increase of the reconstruction error due to the nonconvexity. These results mean that we eliminate the instability of non-negative tensor ring decomposition by our convex formulation.

## 4 Conclusion

We propose *many-body approximation* for tensors, which decomposes tensors by focusing on the relationship between modes represented by an energy-based model. This method approximates and regularizes tensors by ignoring the energy corresponding to certain interactions, which can be viewed as a generalized formulation of mean-field approximation that considers only one-body interactions. Our novel formulation enables us to achieve convex optimization of the model, while the existing approaches based on the low-rank structure are non-convex and empirically unstable with respect to the rank selection. Furthermore, we introduce a way of visualizing activated interactions between modes, called interaction representation. We have demonstrated transformation between our representation and tensor networks, which reveals the nontrivial connection between many-body approximation and the classical tensor low-rank tensor decomposition. We have empirically showed the intuitive model's designability and the better usefulness in tensor completion and approximation tasks compared to baseline methods.

**Limitation** Proposal only works on non-negative tensors. We have examined the effectiveness of LBTC on only traffic datasets.

## Acknowledgments and Disclosure of Funding

This work was supported by RIKEN, Special Postdoctoral Researcher Program (KG), JST, CREST Grant Number JPMJCR22D3, Japan, and JSPS KAKENHI Grant Number JP21H03503 (MS), and JST, CREST Grant Number JPMJCR1913, Japan (YK).

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

# Appendix

## Table of Contents

## A  Projection theory in information geometry

We explain concepts of information geometry used in this study, including natural parameters, expectation parameters, model flatness, and convexity of optimization. In the following discussion, we consider only discrete probability distributions.

$(\theta, \eta)$**-coordinate and geodesics**  In this study, we treat a normalized $D$-order non-negative tensor $\mathcal{P} \in \mathbb{R}_{\geq 0}^{I_1 \times \cdots \times I_D}$ as a discrete probability distribution with $D$ random variables. Let $\mathcal{U}$ be the set of discrete probability distributions with $D$ random variables. The entire space $\mathcal{U}$ is a non-Euclidean space with the Fisher information matrix $\mathbf{G}$ as the metric. This metric measures the distance between two points. In Euclidean space, the shortest path between two points is a straight line. In a non-Euclidean space, such a shortest path is called a geodesic. In the space $\mathcal{U}$, two kinds of geodesics can be introduced: $e$-geodesics and $m$-geodesics. For two points $\mathcal{P}_1, \mathcal{P}_2 \in \mathcal{U}$, $e$- and $m$-geodesics are defined as

$$\left\{ \, \mathcal{R}_t \mid \log \mathcal{R}_t = (1 - t) \log \mathcal{P}_1 + t \log \mathcal{P}_2 - \phi(t) \, \right\}, \quad \left\{ \, \mathcal{R}_t \mid \mathcal{R}_t = (1 - t)\mathcal{P}_1 + t\mathcal{P}_2 \, \right\},$$

respectively, where $0 \leq t \leq 1$ and $\phi(t)$ is a normalization factor to keep $\mathcal{R}_t$ to be a distribution.

We can parameterize distributions $\mathcal{P} \in \mathcal{U}$ by parameters known as natural parameters. In Equation (1), we have described the relationship between a distribution $\mathcal{P}$ and a natural parameter vector $\boldsymbol{\theta} =$

$(\theta_{2,\dots,1}, \dots, \theta_{I_1,\dots,I_D})$. The natural parameter $\boldsymbol{\theta}$ serves as a coordinate system of $\mathcal{U}$, since any distribution in $\mathcal{U}$ is specified by determining $\boldsymbol{\theta}$. Furthermore, we can also specify a distribution $\mathcal{P}$ by its expectation parameter vector $\boldsymbol{\eta} = (\eta_{2,\dots,1}, \dots, \eta_{I_1,\dots,I_D})$, which corresponds to expected values of the distribution and an alternative coordinate system of $\mathcal{U}$. The definition of the expectation parameter $\boldsymbol{\eta}$ is described in Equation (3). $\theta$-coordinates and $\eta$-coordinates are orthogonal with each other, which means that the Fisher information matrix $\mathbf{G}$ has the following property, $\mathbf{G}_{u,v} = \partial\eta_u/\partial\theta_v$ and $(\mathbf{G}^{-1})_{u,v} = \partial\theta_u/\partial\eta_v$. $e$- and $m$-geodesics can also be described using these parameters as follows.

$$\left\{ \boldsymbol{\theta}^t \mid \boldsymbol{\theta}^t = (1-t)\boldsymbol{\theta}^{\mathcal{P}_1} + t\boldsymbol{\theta}^{\mathcal{P}_2} \right\}, \quad \left\{ \boldsymbol{\eta}^t \mid \boldsymbol{\eta}^t = (1-t)\boldsymbol{\eta}^{\mathcal{P}_1} + t\boldsymbol{\eta}^{\mathcal{P}_2} \right\},$$

where $\boldsymbol{\theta}^{\mathcal{P}}$ and $\boldsymbol{\eta}^{\mathcal{P}}$ are $\theta$- and $\eta$-coordinate of a distribution $\mathcal{P} \in \mathcal{U}$.

**Flatness and projections**  A subspace is called $e$-flat when any $e$-geodesic connecting two points in a subspace is included in the subspace. The vertical descent of an $m$-geodesic from a point $\mathcal{P} \in \mathcal{U}$ onto $e$-flat subspace $\mathcal{B}_e$ is called $m$-projection. Similarly, $e$-projection is obtained when we replace all $e$ with $m$ and $m$ with $e$. The flatness of subspaces guarantees the uniqueness of the projection destination. The projection destination $\overline{\mathcal{P}}$ or $\tilde{\mathcal{P}}$ obtained by $m$- or $e$-projection onto $\mathcal{B}_e$ or $\mathcal{B}_m$ minimizes the following KL divergence,

$$\overline{\mathcal{P}} = \arg\min_{\mathcal{Q} \in \mathcal{B}_e} D_{KL}(\mathcal{P}, \mathcal{Q}), \quad \tilde{\mathcal{P}} = \arg\min_{\mathcal{Q} \in \mathcal{B}_m} D_{KL}(\mathcal{Q}, \mathcal{P}).$$

The KL divergence from discrete distributions $\mathcal{P} \in \mathcal{U}$ to $\mathcal{Q} \in \mathcal{U}$ is given as

$$D_{KL}(\mathcal{P}, \mathcal{Q}) = \sum_{i_1=1}^{I_1} \cdots \sum_{i_D=1}^{I_D} \mathcal{P}_{i_1,\dots,i_D} \log \frac{\mathcal{P}_{i_1,\dots,i_D}}{\mathcal{Q}_{i_1,\dots,i_D}}. \tag{13}$$

A subspace with some of its natural parameters fixed at $0$ is $e$-flat [1, Chapter 2], which is obvious from the definition of $e$-flatness. The proposed many-body approximation performs $m$-projection onto the subspace $\mathcal{B} \subset \mathcal{U}$ with some natural parameters fixed to be $0$. From this linear constraint, we know that $\mathcal{B}$ is $e$-flat. Therefore, the optimal solution of the many-body approximation is always unique. When a space is $e$-flat and $m$-flat at the same time, we say that the space is dually-flat. The set of discrete probability distributions $\mathcal{U}$ is dually-flat.

**Natural gradient method**  $e(m)$-flatness guarantees that cost functions to be optimized in Equation (13) are convex. Therefore, $m(e)$-projection onto an $e(m)$-flat subspace can be implemented by a gradient method using a second-order gradient. This second-order gradient method is known as the natural gradient method [42]. The Fisher information matrix $\mathbf{G}$ appears by second-order differentiation of the KL divergence (see Equation (5)). We can perform fast optimization using the update formula in Equation (6), using the inverse of the Fisher information matrix.

**Examples for Möbius function**  In the proposed method, we need to transform the distribution $\mathcal{P} \in \mathbb{R}^{I_1 \times \cdots \times I_D}$ with $\boldsymbol{\theta}$ and $\boldsymbol{\eta}$ using the Möbius function, defined in Section 2.1. We provide examples here. In Equation (2), The Möbius function is used to find the natural parameter $\boldsymbol{\theta}$ from a distribution $\mathcal{P}$. For example, if $D = 2, 3$, it holds that

$$\theta_{i_1,i_2} = \log \mathcal{P}_{i_1,i_2} - \log \mathcal{P}_{i_1-1,i_2} - \log \mathcal{P}_{i_1,i_2-1} + \log \mathcal{P}_{i_1-1,i_2-1},$$
$$\theta_{i_1,i_2,i_3} = \log \mathcal{P}_{i_1,i_2,i_3} - \log \mathcal{P}_{i_1-1,i_2,i_3} - \log \mathcal{P}_{i_1,i_2-1,i_3} - \log \mathcal{P}_{i_1,i_2,i_3-1}$$
$$+ \log \mathcal{P}_{i_1-1,i_2-1,i_3} + \log \mathcal{P}_{i_1-1,i_2,i_3-1} + \log \mathcal{P}_{i_1,i_2-1,i_3-1} - \log \mathcal{P}_{i_1-1,i_2-1,i_3-1},$$

where we assume $\mathcal{P}_{0,i_2} = \mathcal{P}_{i_1,0} = 1$ and $\mathcal{P}_{i_1,i_2,0} = \mathcal{P}_{i_1,0,i_3} = \mathcal{P}_{0,i_2,i_3} = 1$. To identify the value of $\theta_{i_1,\dots,i_d}$, we need only $\mathcal{P}_{i'_1,\dots,i'_d}$ with $(i'_1, \dots, i'_d) \in \{i_1 - 1, i_1\} \times \{i_2 - 1, i_2\} \times \cdots \times \{i_d - 1, i_d\}$. In the same way, using Equation (3), we can find the distribution $\mathcal{P}$ by the expectation parameter $\boldsymbol{\eta}$. For example, if $D = 2, 3$, it holds that

$$\mathcal{P}_{i_1,i_2} = \eta_{i_1,i_2} - \eta_{i_1+1,i_2} - \eta_{i_1,i_2+1} + \eta_{i_1+1,i_2+1},$$
$$\mathcal{P}_{i_1,i_2,i_3} = \eta_{i_1,i_2,i_3} - \eta_{i_1+1,i_2,i_3} - \eta_{i_1,i_2+1,i_3} - \eta_{i_1,i_2,i_3+1}$$
$$+ \eta_{i_1+1,i_2+1,i_3} + \eta_{i_1+1,i_2,i_3+1} + \eta_{i_1,i_2+1,i_3+1} - \eta_{i_1+1,i_2+1,i_3+1},$$

where we assume $\eta_{I_1+1,i_2} = \eta_{i_1,I_2+1} = 0$ and $\eta_{I_1+1,i_2,i_3} = \eta_{i_1,I_2+1,i_3} = \eta_{i_1,i_2,I_3+1} = 0$.

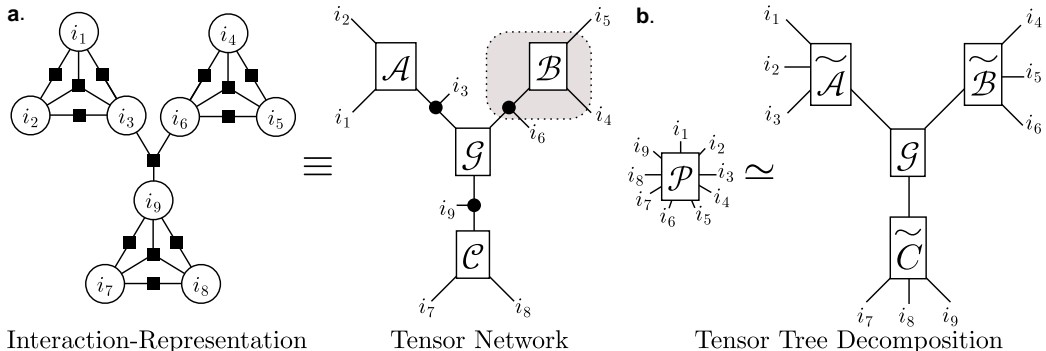

Interaction-Representation      Tensor Network      Tensor Tree Decomposition

Figure 5: (**a**) Interaction representation corresponding to Equation (14) and its transformed tensor network for $D = 9$. (**b**) Tensor network of a variant of tensor tree decomposition.

## B   Formal definitions

We provide give formal definitions of Legendre decomposition [35] as follows:

**Definition 1.** *For a normalized non-negative tensor $\mathcal{P}$ and a set of indices $B$, Legendre decomposition of $\mathcal{P}$ is $\mathcal{Q}$ defined as*

$$\overline{\mathcal{Q}} = \arg\min_{\mathcal{Q};\mathcal{Q}\in\boldsymbol{\mathcal{B}}} D_{KL}(\mathcal{P}, \mathcal{Q})$$

*where $\boldsymbol{\mathcal{B}} = \{\mathcal{Q} \mid \theta_{i_1,\dots,i_D} = 0 \text{ if } (i_1,\dots,i_D) \notin B \text{ for } \theta\text{-parameters of } \mathcal{Q}\}$.*

As in the following definition, a particular case of the Legendre decomposition is the many-body decomposition.

**Definition 2.** *For a normalized non-negative tensor $\mathcal{P}$ and a natural number $m$, we define its $m$-body approximation $\mathcal{P}^{\leq m}$ as*

$$\overline{\mathcal{Q}} = \arg\min_{\mathcal{Q};\mathcal{Q}\in\boldsymbol{\mathcal{B}}} D_{KL}(\mathcal{P}, \mathcal{Q})$$

*where $\boldsymbol{\mathcal{B}}$ used in $B$ contains all the indices of $n(\leq m)$-body parameters.*

## C   Theoretical remarks

### C.1   Convexity and uniqueness of many-body approximation

It is widely known that the maximum likelihood estimation of ordinary Boltzmann machines without hidden variables is a convex problem. Since we optimize the KL divergence, the proposed many-body approximation is also a maximum likelihood estimation that approximates a non-negative tensor, which is regarded as an empirical distribution, by an extended Boltzmann machine without hidden variables, as described in Section 2.2. As with ordinary Boltzmann machines, the maximum likelihood estimation of extended Boltzmann machines can be globally optimized. Theorem 1 is a general proof of this using information geometry.

**Theorem 1.** *The solution of many-body approximation is always unique, and the objective function of many-body approximation is convex.*

*Proof.* As Definition 2 shows, the objective function of the proposed many-body approximation is the KL divergence from an empirical distribution (a given normalized tensor) to a subspace $\boldsymbol{\mathcal{B}}$. We can immediately prove that a subspace $\boldsymbol{\mathcal{B}}$ is $e$-flat from the definition of the flatness [1, Chapter 2] for any $B$. The KL divergence minimization from a given distribution to $e$-flat space is called $m$-projection (see the second paragraph in Section A). The $m$-projection onto $e$-flat subspace is known to be convex and the destination is unique (see the third paragraph in Section A). Thus, the optimal solution of the many-body approximation is always unique, and the objective function is convex.   □

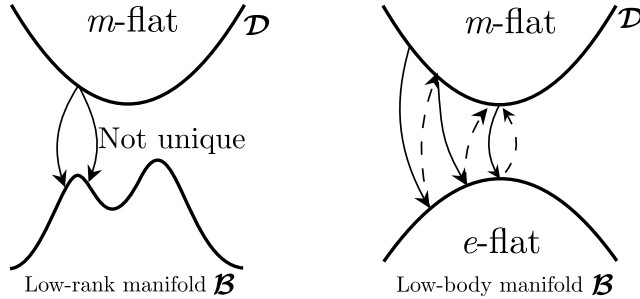

Figure 6: The $em$-algorithm. Dashed(Solid) arrows are $e(m)$-projection.

## C.2 Another example: Tensor tree decomposition

As seen in Section 2.4, cyclic two-body approximation coincides with tensor ring decomposition with a constraint. In this section, we additionally provide another example of correspondence between many-body approximation and the existing low-rank approximation. For $D = 9$, let us consider three-body and two-body interactions among $(i_1, i_2, i_3)$, $(i_4, i_5, i_6)$, and $(i_7, i_8, i_9)$ and three-body approximation among $(i_3, i_6, i_9)$. We provide the interaction representation of the target energy function in Figure 5(**a**). In this approximation, the decomposed tensor can be described as

$$\mathcal{P}_{i_1,\dots,i_9} = \mathcal{A}_{i_1,i_2,i_3}\mathcal{B}_{i_4,i_5,i_6}\mathcal{C}_{i_7,i_8,i_9}\mathcal{G}_{i_3,i_6,i_9}. \tag{14}$$

Its converted tensor network is described in Figure 5(**a**). It becomes a tensor network of tensor tree decomposition in Figure 5(**b**) when the region enclosed by the dotted line is replaced with a new tensor (shown with tilde) by coarse-graining. Such tensor tree decomposition is used in generative modeling [43], computational chemistry [48] and quantum many-body physics [50].

## C.3 Reducing computational complexity by reshaping tensor

As described in Section 2.1.1, the computational complexity of many-body approximation is $\mathcal{O}(\gamma|B|^3)$, where $\gamma$ is the number of iterations, because the overall complexity is dominated by the update of $\theta$, which includes matrix inversion of $\mathbf{G}$ and the complexity of computing the inverse of an $n \times n$ matrix is $\mathcal{O}(n^3)$. This complexity can be reduced if we reshape tensors so that the size of each mode becomes small. For example, let us consider a third-order tensor whose size is $(J^2, J^2, J^2)$ and its cyclic two-body approximation. In this case, the time complexity is $\mathcal{O}(\gamma J^{12})$ since it holds that $|B| = \mathcal{O}(J^4)$ (see Equation (12)). By contrast, if we reshape the input tensor to a sixth-order tensor whose size is $(J, J, J, J, J, J)$, the time complexity becomes $\mathcal{O}(\gamma J^6)$ since it holds that $|B| = \mathcal{O}(J^2)$.

This technique of reshaping a tensor into a larger-order tensor is widely used practically in various methods based on tensor networks, such as tensor ring decomposition [47]. We use this technique in the experiment in Section 3.3 to handle large tensors in the both of proposal and baselines.

## C.4 Information geometric view of LBTC

Traditional methods for tensor completion are based on the following iteration, called $em$-algorithm [51].

$$e\text{-step: } \mathcal{P}_\Omega \leftarrow \mathcal{T}_\Omega, \qquad m\text{-step}: \mathcal{P} \leftarrow \texttt{low-rank-approximation}(\mathcal{P}),$$

with an initialized tensor $\mathcal{P}$, where $\Omega$ is the set of observed indices of a given tensor $\mathcal{T}$ with missing values. From the information geometric view, $m$-step does $m$-projection onto the model manifold $\mathcal{B}$ consisting of low-rank tensors, and $e$-step does $e$-projection onto the data manifold $\mathcal{D} \equiv \{\mathcal{P} \mid \mathcal{P}_\Omega = \mathcal{T}_\Omega\}$. It is straightforward to prove that $e$-step finds a point $\overline{\mathcal{P}} = \arg\min_{\overline{\mathcal{P}} \in \mathcal{D}} D_{KL}(\overline{\mathcal{P}}, \mathcal{P})$ for the obtained tensor $\mathcal{P}$ by the previous $m$-step [52]. The iteration of $e$- and $m$-step finds the closest point between two manifolds $\mathcal{B}$ and $\mathcal{D}$. Since we can immediately prove that $t\mathcal{Q}_1 + (1-t)\mathcal{Q}_2$ belongs to $\mathcal{D}$ for any $\mathcal{Q}_1, \mathcal{Q}_2 \in \mathcal{D}$ and $0 \leq t \leq 1$, the data manifold $\mathcal{D}$

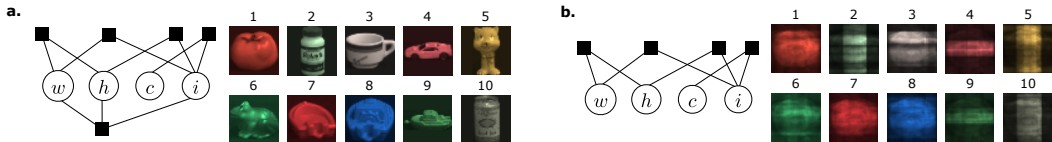

Figure 7: (**a**) Many-body approximation with three-body interaction among width, height, and index image can reconstruct the shapes of each image. (**b**) Without the interaction, many-body approximation cannot reconstruct the shapes of each image.

is $m$-flat, which guarantees the uniqueness of the $e$-step. See the definition of $m$-flat in Section A. However, since low-rank subspace is not $e$-flat, that $m$-projection is not unique. LBTC conducts tensor completion by replacing low-rank approximation in $m$-step with many-body approximation. Since the model manifold of many-body approximation is always $e$-flat and data manifold is $m$-flat, each step in the algorithm is unique.

# D    Related work

As discussed in Section 1, we assume low-rankness in the structure of the tensor in conventional decompositions. However, since tensors have many modes, there are too many kinds of low-rank structures. Examples include, the CP decomposition [15], which decomposes a tensor by the sum of Kronecker products of vectors; the Tucker decomposition [37], which decomposes a tensor by a core tensor and multiple matrices; the tensor train decomposition [29], which decomposes a tensor by multiple core tensors; and tensor-ring decomposition [41], which assume periodic structure on tensor train decomposition are well known and popularly used in machine learning.

However, in general, it is difficult to find out the low-rank structure of a given data set. Therefore, it is difficult to say how to choose a low-rank decomposition model from those options. To alleviate this difficulty in model selection and provide intuitive modeling, we have introduced the concept of body and formulated a decomposition that does not rely on ranks.

# E    Additional numerical results

## E.1    Additional results for COIL dataset

In Section 3.1, we see that interactions related to color control color-richness in the reconstruction. We provide additional experimental results in Figure 7 to see the role of the interaction among width, height, and image index. The figure shows the influence of the interaction for reconstruction. While Figure 7(**b**), without the interaction, could not capture the shapes of images, Figure 7(**a**) reconstruct each shape properly. Based on these results, it is reasonable that the interaction is responsible for these shapes.

## E.2    Additional results for LBTC

We provide the reconstructions by the experiments in Section 3.2 in Figure 8 with ground truth.

In traditional low-rank approximation, tuning hyper-parameters based on prior knowledge is difficult because its semantics are unclear. By contrast, the interactions in many-body approximation can be tuned easily by considering the expected correlation among modes of a given tensor. To support this claim, we also confirm that LBTC scores get better when we activate interactions between modes that are expected to be correlated based on prior knowledge. In this experiment, we use the same datasets in Section 3.2. We prepare two set of interactions, Interactions A and B, as shown on the left in Figure 9. In Interaction A, we activate interaction between modes such as (date, hour) and (hour, minute, lane), which we expect to be correlated. In Interaction B, we activate interaction between modes such as (date, minute) and (minute, lane), which we do not expect to be correlated. For a fair comparison, we chose these interactions so that the number of parameters used in Interactions A and B are as close as possible, where the number of parameters for Interaction A was 1847, and the number of parameters for Interaction B was 1619. As seen in Table 2, LBTC with Interaction A always has a better score than Interaction B, which implies that many-body approximation has better

performance when we choose interactions among correlated modes. We conducted the experiment 10 times and reported mean values with standard error. We also visualize the reconstruction in Figure 9 to show that LBTC with Interaction A can complete tensors more accurately than that with Interaction B.

As seen in Figure 14 and Section G.2, existing methods require hyper-parameters that are difficult to tune since we usually cannot find any correspondence between these values and prior knowledge about the given tensor. In contrast, as demonstrated in this section, our energy-based model is intuitively tunable that can skip time-consuming hyper-parameter tuning.

Table 2: Dependency of recovery fit score of LBTC on a choice of interaction.

|  | **Scenario 1** | **Scenario 2** | **Scenario 3** |
|---|---|---|---|
| Interaction A | $0.900 \pm 0.00125$ | $0.889 \pm 0.000632$ | $0.864 \pm 0.001264$ |
| Interaction B | $0.820 \pm 0.00143$ | $0.819 \pm 0.000699$ | $0.805 \pm 0.000527$ |

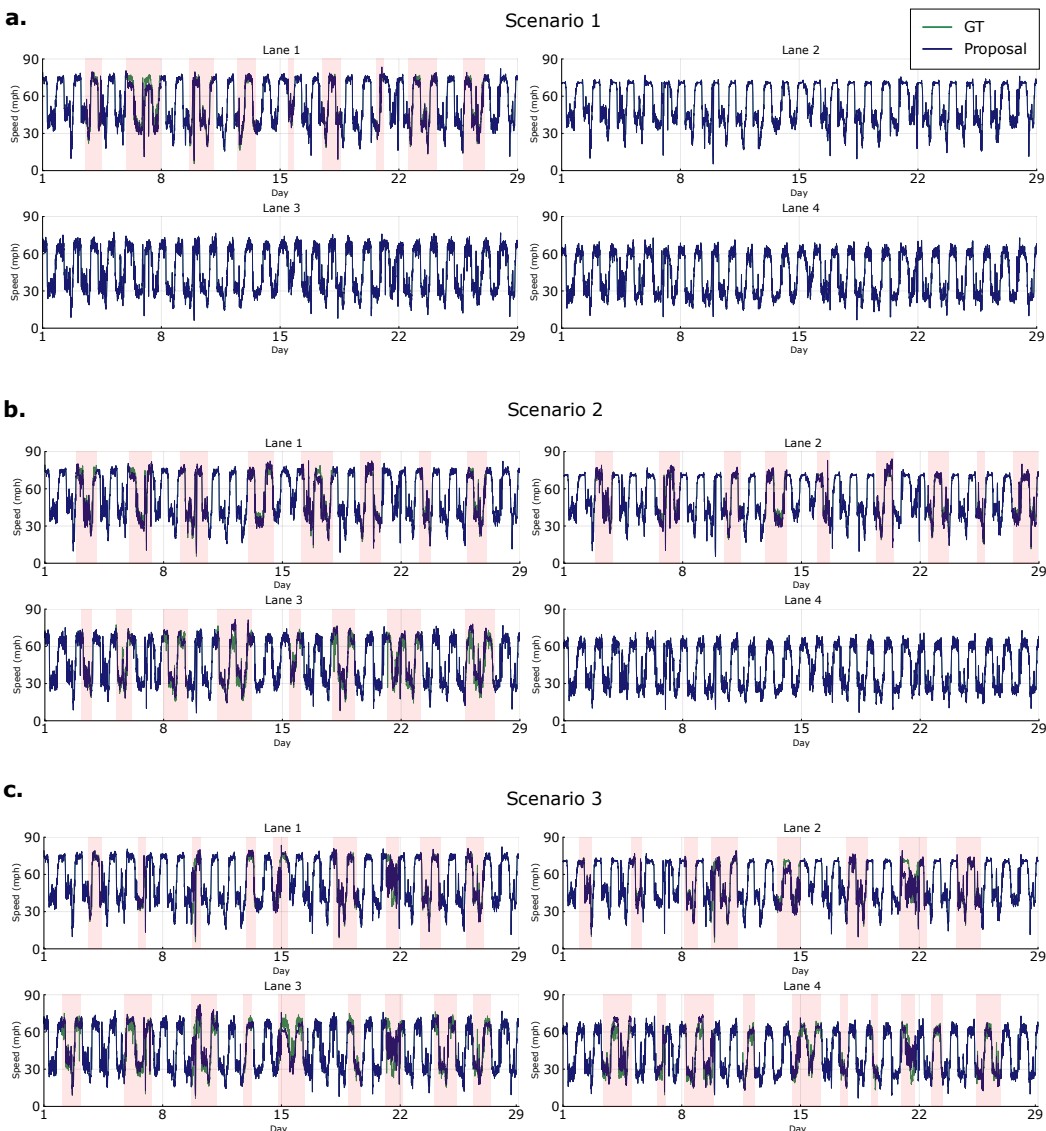

Figure 8: Ground truth and reconstruction by LBTC in the experiment in Section 3.2. Filled areas are missing parts, and blue lines are estimated values by the proposal. Green lines are ground truth.

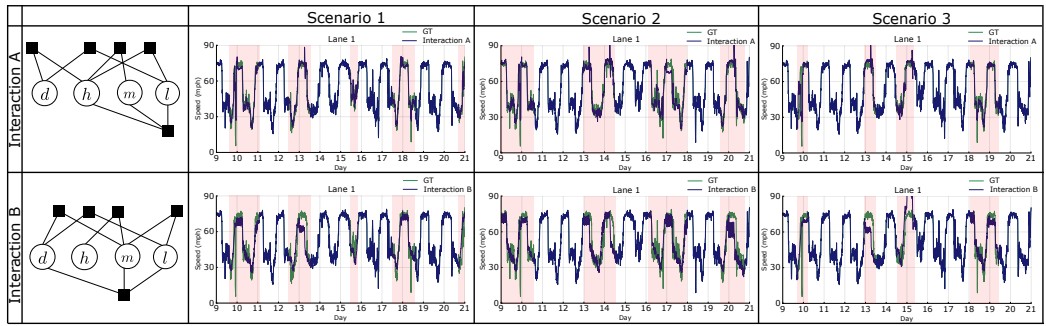

Figure 9: Interaction dependency of LBTC. Filled areas are missing parts, and blue lines are estimated values by the proposal with Interactions A and B, respectively. Green lines are ground truth. Due to space limitations, we show the reconstruction of Lane 1 from day 9 to 20, where many missing parts exists.

### E.3 Additional results for comparison with ring decomposition

We additionally provide Figure 10, which shows running time compassion in the experiment in Section 3.3. As described in the main text, we repeat five times each baseline method with random initialization while the proposal runs only once. We compare the total running time of them. As seen in Figure 10, for both synthetic and real datasets, the proposed method is always faster than baselines, keeping the competitive relative errors. The flatness of solution space enables many-body approximation to be solved with the Newton method. In contrast, NTR is solved by a first-order derivative. The quadratic convergence of the Newton method is well known, and it requires only a few iterations to find the best solution in the flat solution space $\mathcal{B}_{\mathrm{cyc}}$. That is the reason why the proposed method is fast and stable.

### E.4 Comparison with traditional low-rank approximation

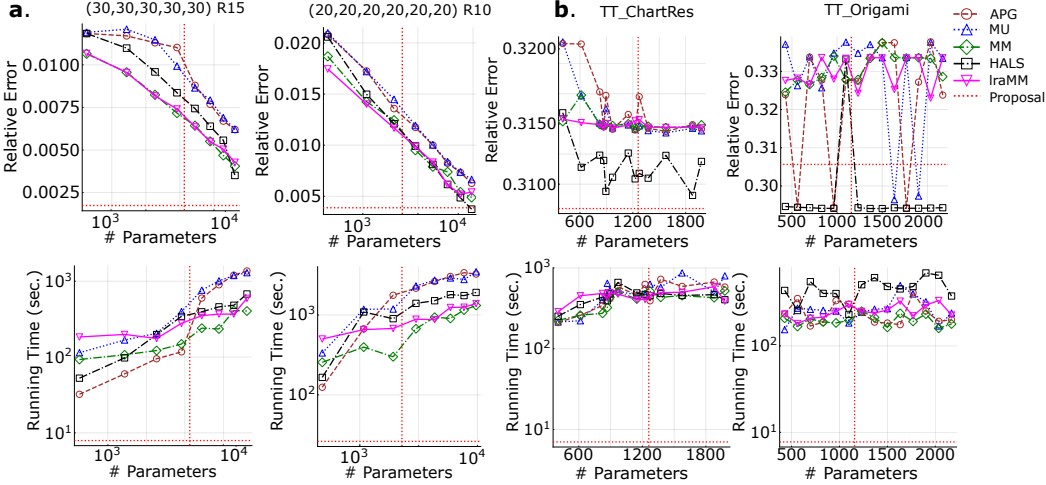

Figure 10: Comparison of reconstruction capabilities (top) and computational speed (bottom) of cyclic two-body approximation and tensor ring decomposition for synthetic low-tensor ring datasets (**a**) and real datasets (**b**). The vertical red dotted line is $|\mathcal{B}|$ (See Equation (12)).

In low-rank approximation, we often gradually increase the rank to enlarge model capability. In the same way, if we have no prior or domain knowledge about a tensor, we can gradually increase $m$ of $m$-body approximation and get more accurate reconstruction. To demonstrate that, we conducted additional experiments with the same dataset in Section 3.2. We compared the reconstruction ability

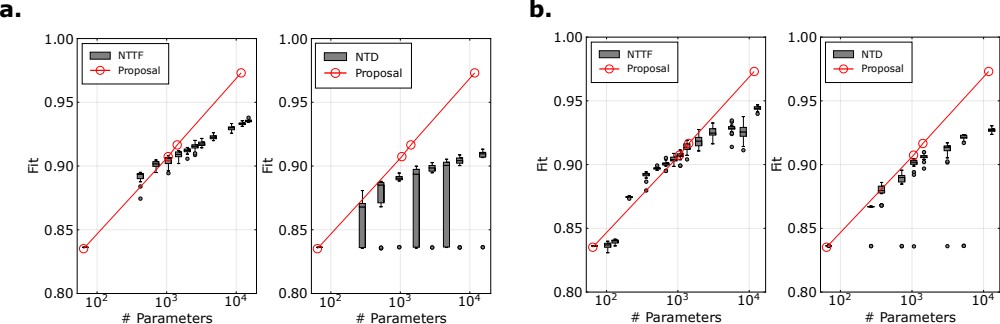

Figure 11: We compared the fit score of proposal $\mathcal{P}^{\leq 1}, \mathcal{P}^{\mathrm{cyc}}, \mathcal{P}^{\leq 2}, \mathcal{P}^{\leq 3}$ with baselines. (**a**) The ranks for baselines assumed as $(r, \dots, r)$ with changing the value of $r$. (**b**) The ranks for baselines are tuned taking the tensor into size account.

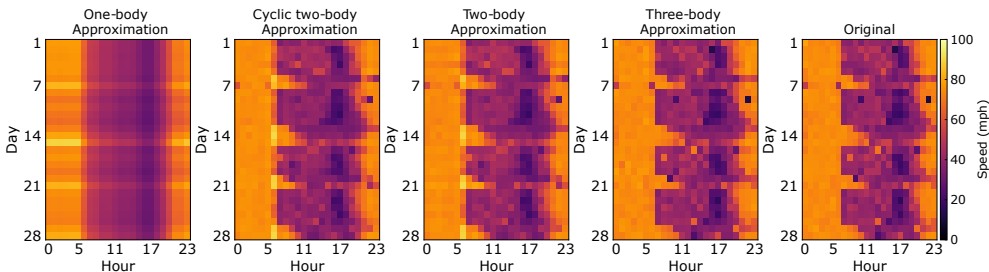

Figure 12: Heatmaps for reconstructed tensor $\mathcal{P}^{\leq 1}, \mathcal{P}^{\mathrm{cyc}}, \mathcal{P}^{\leq 2}, \mathcal{P}^{\leq 3}$ and input tensor $\mathcal{P}$. We visualize the traffic on only the first lane at 00 minutes every hour, which corresponds to $\mathcal{P}[:, :, 1, 1]$.

of the proposed $m$-body approximation with non-negative tucker decomposition (NTD) [44] and non-negative tensor train decomposition (NTTF) [34].

In many applications based on matrix and tensor factorization, their objective functions are often the fit score (RMSE) and they try to find good representations to solve the task by optimizing the objective. Therefore, for fair evaluation without depending on specific applications, we examine the performance by fit score.

For $m$-body approximation, we gradually increased the order of the interaction by changing $m$ to $1, 2, 3$. We ran the proposal only once since the proposal does not depend on initial values. For baselines, we gradually enlarged model capacity by increasing the target rank. We repeated each baseline 20 times with random initialization. We defined the Tucker and train ranks for the baseline method in two ways. First, we assumed that all values in ranks are the same – that is, $(r, \dots, r)$ – and increase the value of $r$. However, since the size of an input tensor is $(28, 24, 12, 4)$ and each mode size is not unique, it can be unnatural rank setting, which may cause worse RMSE. To avoid that, we also adjusted target rank by taking the tensor size into account. In particular, the rank is set to be $(1,1,1,1), (6,2,2,1), (6,5,2,1), (8,5,4,2), (10,7,4,2), (12,9,4,2), (16,12,6,2), (18,15,8,2), (20,18,11,3)$ for NTD and $(1,1,1), (1,2,1), (1,2,2), (2,2,2), (4,2,2), (6,2,2), (6,3,2), (8,3,2), (10,3,2), (10,4,2), (15,4,2), (20,5,2), (20,8,2), (30,10,2), (30,15,8)$ for NTTF.

We evaluated the approximation performance with the fit score $1 - \|\mathcal{P} - \overline{\mathcal{P}}\|_F / \|\mathcal{P}\|_F$ for the input $\mathcal{P}$ and the reconstructed tensor $\overline{\mathcal{P}}$. We show these results in Figure 11. While baselines are often trapped in a worse local minimum, the proposal reconstructs the original tensor better. We also show the reconstruction by the proposed method as heatmaps in Figure 12. Even one-body approximation is able to reconstruct the rush hour around 17:00. When the number of interactions increases, we can observe the trend that the time of the busy hours on weekends are different from those on weekdays more clearly.

To see the effectiveness of proposal for data mining, we also plotted two-body interaction between day and lane extracted by the two-body approximation of the above traffic data. This is a matrix

$\mathbf{X}^{(d,l)} \in \mathbb{R}^{28 \times 4}$. As seen in Figure 13, the value in the first column is always the largest and the value in the fourth column is always the smallest. From this result, we can interpret that the data corresponding to $\mathcal{P}[:, :, :, 1]$ were taken in the overtaking lane and the data for $\mathcal{P}[:, :, :, 4]$ were taken in the overtaken lane.

# F  Dataset details

We describe the details of each dataset below.

## F.1  For experiment with RGB images

We downloaded the COIL-100 dataset from the official website.[1] We used 10 images with object numbers 4, 5, 7, 10, 17, 23, 28, 38, 42, and 78. This selection aimed to include both monochromatic and multi-colored objects so that we can see how the proposed method changes the reconstruction if we deactivate the interactions related to color. We reshaped the size of each image to $40 \times 40$.

## F.2  For experiments with LRTC

We downloaded traffic speed records in District 7, Los Angeles County, from PeMS.[2] The data are in the public domain. The whole period of the data lasts for 28 days from February 4 to March 3, 2013. By measuring the speed of vehicles in four lanes at a certain point on the highway 12 times at five-minute intervals, a $12 \times 4$ matrix is generated per hour. Twenty-four of these matrices are generated per day, so the dataset is a $24 \times 12 \times 4$ tensor. Further, since we used these tensors for 28 days, the size of the resulting tensor is $28 \times 24 \times 12 \times 4$. We artificially generated defects in this tensor. As practical situations, we assumed that a disorder of a lane's speedometer causes random and continuous deficiencies. We prepared three patterns of missing cases: a case with a disorder in the speedometer in Lane 1 (Scenario 1), a case with disorders in the speedometers in Lanes 1, 2, and 3 (Scenario 2), and a case with disorders in all speedometers (Scenario 3). The missing rates for Scenarios 1, 2, and 3 were 9 percent, 27 percent, and 34 percent, respectively. The source code for imparting deficits are available in the Supplementary Materials.

## F.3  For comparison with non-negative tensor ring decomposition

**Synthetic datasets**   We created $D$ core tensors with the size $(R, I, R)$ by sampling from uniform distribution. A tensor with the size $I^D$ and the tensor ring rank $(R, \ldots, R)$ was then obtained by the product of these $D$ tensors. Results for $R = 15, D = 5, I = 30$ are shown in the left column in Figure 4, and those for $R = 10, D = 6, I = 20$ in the right column in Figure 4. For all experiments on synthetic datasets, we change the target ring-rank as $(r, \ldots, r)$ for $r = 2, 3, \ldots, 9$ for baseline methods.

**Real datasets**   TT_ChartRes is originally a $(736, 736, 31)$ tensor, produced from TokyoTech 31-band Hyperspectral Image Dataset. We downloaded ChartRes.mat from the official website[3] and reshaped the tensor as $(23, 8, 4, 23, 8, 4, 31)$. For baseline methods, we chose the target ring-rank as $(2, 2, 2, 2, 2, 2, 2)$ $(2, 2, 2, 2, 2, 2, 5)$, $(2, 2, 2, 2, 2, 2, 8)$, $(3, 2, 2, 3, 2, 2, 5)$, $(2, 2, 2, 2, 2, 2, 9)$, $(3, 2, 2, 3, 2, 2, 6)$, $(4, 2, 2, 2, 2, 2, 6)$, $(3, 2, 2, 4, 2, 2, 8)$, $(3, 2, 2, 3, 2, 2, 9)$, $(3, 2, 2, 3, 2, 2, 10)$, $(3, 2, 2, 3, 2, 2, 12)$, $(3, 2, 2, 3, 2, 2, 15)$, or $(3, 2, 2, 3, 2, 2, 16)$.   TT_Origami is originally $(512, 512, 59)$ tensor, which is produced from TokyoTech 59-band Hyperspectral Image Dataset. We downloaded Origami.mat from the official website.[4] In TT_Origami, 0.0016 percent of elements were negative, hence we preprocessed all elements of TT_Origami by subtracting $-0.000764$, the smallest value in TT_Origami, to make all elements non-negative. We reshaped the tensor as $(8, 8, 8, 8, 8, 8, 59)$. For baseline methods, we chose the target ring-rank as $(2, 2, 2, 2, 2, 2, r)$ for $r = 2, 3, \ldots, 15$. These reshaping reduces the computational complexity as described in Section C.3 to complete the experiment in a reasonable time.

---

[1] https://www1.cs.columbia.edu/CAVE/software/softlib/coil-100.php
[2] http://pems.dot.ca.gov/
[3] http://www.ok.sc.e.titech.ac.jp/res/MSI/MSIdata31.html
[4] http://www.ok.sc.e.titech.ac.jp/res/MSI/MSIdata59.html

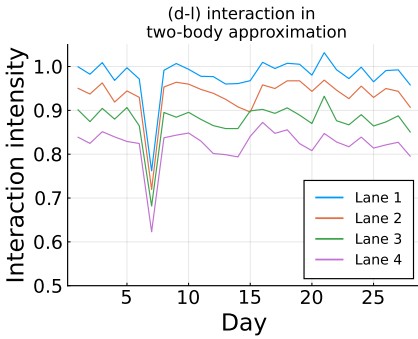

Figure 13: The obtained interaction between day and lane in two-body approximation.

## G Implementation detail

We describe the implementation details of methods in the following. All methods are implemented in Julia 1.8.

**Environment** Experiments were conducted on Ubuntu 20.04.1 with a single core of 2.1GHz Intel Xeon CPU Gold 5218 and 128GB of memory.

### G.1 Proposed method

**Many-body approximation** We use a natural gradient method for many-body approximation. The natural gradient method uses the inverse of the Fisher information matrix to perform second-order optimization in a non-Euclidean space. For non-normalized tensors, we conducted the following procedure. First, we computed the total sum of elements of an input tensor. Then, we normalized the tensor. After that, we conducted Legendre decomposition for the normalized tensor. Finally, we obtained the product of the result of the previous step and the total sum we computed initially. That procedure was justified by the general property of the KL divergence $\lambda D_{KL}(\mathcal{P}, \mathcal{Q}) = D_{KL}(\lambda \mathcal{P}, \lambda \mathcal{Q})$, for any $\lambda \in \mathbb{R}_{>0}$. The termination criterion is the same as the original implementation of Legendre decomposition by [36]; that is, it terminates if $||\boldsymbol{\eta}_t^B - \hat{\boldsymbol{\eta}}^B|| < 10^{-5}$, where $\boldsymbol{\eta}_t^B$ is the expectation parameters on $t$-th step and $\hat{\boldsymbol{\eta}}^B$ is the expectation parameters of an input tensor, which are defined in Section 2.1.1. The overall procedure is described in Algorithm 2. Note that this algorithm is based on Legendre decomposition by [36].

**LBTC** The overall procedure is described in Algorithm 1 with our tolerance threshold. We randomly initialized each missing element in $\mathcal{P}^{(t=1)}$ by Gaussian distribution with mean 50 and variance 10.

### G.2 Baselines

**Baseline methods for tensor completion** We implemented baseline methods for tensor completion by translating pseudo codes in original papers into Julia code. As baseline methods, we used HaLRTC, SiLRTC [21], SiLRTC-TT [2], and PTRCRW [38]. SiLRTC and HaLRTC are well-known classical methods based on HOSVD. SiLRTC-TT is based on tensor train decomposition. PTRCRW is based on tensor-ring decomposition and is recognized as the state of the art. HaLRTC requires a real value $\rho$ and a weight vector $\boldsymbol{\alpha}$. We used the default setting on $\boldsymbol{\alpha}$ described in the original paper and tuned the value $\rho$ as $\rho = 10^{-5}, 10^{-4}, \ldots, 10^2$. SiLRTC requires a hyperparameter $\gamma_d$ for $d = 1, 2, 3, 4$. This is the threshold of truncated SVD for the mode-$d$ matricization of the input tensor. Following the original paper, we assumed $\tau = \tau_1 = \cdots = \tau_4$ and we tune $\tau$ in the range of $\{10^{-2}, 10^{-1}, \ldots, 10^4\}$. SiLRTC-TT requires a weight vector $\boldsymbol{\alpha}$ and a real number $f$. We set the value of $\boldsymbol{\alpha}$ as the default setting described in the original paper. We tune the value of $f$ in the range of $\{0.001, 0.01, 0.025, 0.1, 0.25, 1\}$. PTRC-RW requires two weight vectors, $\boldsymbol{\alpha}$ and $\boldsymbol{\beta}$, step length $d$, and tensor ring rank as hyper parameters. Following the original paper, we used the default setting for $\boldsymbol{\alpha}$, $\boldsymbol{\beta}$, and $d$. Then we tuned the tensor ring rank $(R, R, R, R)$ within the range $R \in \{1, 2, \ldots, 6\}$, assuming that it has the same values in each element, which is also assumed in the original paper.

---

**Algorithm 2:** Many-body approximation

---

MANYBODYAPPROXIMATION($\mathcal{T}$, $B$)

    $s \leftarrow$ Total sum of $\mathcal{T}$.

    Obtain normalized input tensor $\mathcal{P} \leftarrow \mathcal{T}./s$          // "./" denotes element-wise division

    Compute $\hat{\boldsymbol{\eta}}$ of $\mathcal{P}$ using Equation (3).

    Initialize $\boldsymbol{\theta}_{t=1}^B$                          // e.g. $\theta_b = 0$ for all $b \in B$

    $t \leftarrow 1$

    **repeat**

        Compute $\mathcal{Q}_t$ using the current parameter $\boldsymbol{\theta}_t^B$ with Equation (1).

        Compute $\boldsymbol{\eta}_t^B$ from $\mathcal{Q}_t$ using Equation (3).

        Compute the inverse of the Fisher information matrix $\mathbf{G}$ using Equation (5).

        $\boldsymbol{\theta}_{t+1}^B \leftarrow \boldsymbol{\theta}_t^B - \mathbf{G}^{-1}(\boldsymbol{\eta}_t^B - \hat{\boldsymbol{\eta}}^B)$

        $t \leftarrow t + 1$

    **until** $\|\boldsymbol{\eta}_t^B - \hat{\boldsymbol{\eta}}^B\| < \epsilon$           // We set $\epsilon = 10^{-5}$ in our implementation;

    $\overline{\mathcal{T}} \leftarrow \mathcal{Q}_t .* s$                 // ".*" denotes element-wise multiplication

    **return** $\overline{\mathcal{T}}$

---

Since PTRC-RW is not a convex method, we repeated it 10 times for each $R$ with randomly sampled initial values by Gaussian distribution with mean 50 and variance 10. We provide Figure 14, which shows the sensitivity of the tensor completion performance with respect to changes of the above hyper-parameters. All of the above baseline methods include iterative procedures. These iterations were stopped when the relative error between the previous and the current iteration is less than $10^{-5}$ or when the iteration count exceeds 1500 iterations.

**Baseline methods for tensor ring decomposition** We implemented baseline methods for tensor ring decomposition by translating MATLAB code provided by the authors into Julia code for fair comparison. As we can see from their original papers, NTR-APG, NTR-HALS, NTR-MU, NTR-MM, and NTR-lraMM have an inner and outer loop to find a local solution. We repeated the inner loop 100 times. We stopped the outer loop when the difference between the relative error of the previous and the current iteration is less than $10^{-4}$. NTR-MM and NTR-lraMM require a diagonal parameter matrix $\Xi$. We define $\Xi = \omega I$ where $I$ is an identical matrix and $\omega = 0.1$. The NTR-lraMM method performs low-rank approximation to the matrix obtained by mode expansion of an input tensor. The target rank was set to be 20. This setting is the default setting in the provided code. The initial positions of baseline methods were sampled from uniform distribution on $(0, 1)$.

**Baseline methods for low-rank approximation** We implemented NNTF by translating the pseudo-code in the original paper [34] into Julia code. The algorithm requires NMF in each iteration. For that, we also implement the most standard method, multiplicative update rules, following the paper [46]. We use default values of hyper parameters of `sklearn` [49] for NMF. For NTD, we translate `TensorLy` implementation [45] to Julia code, whose cost function is the Frobenius norm between input and output.

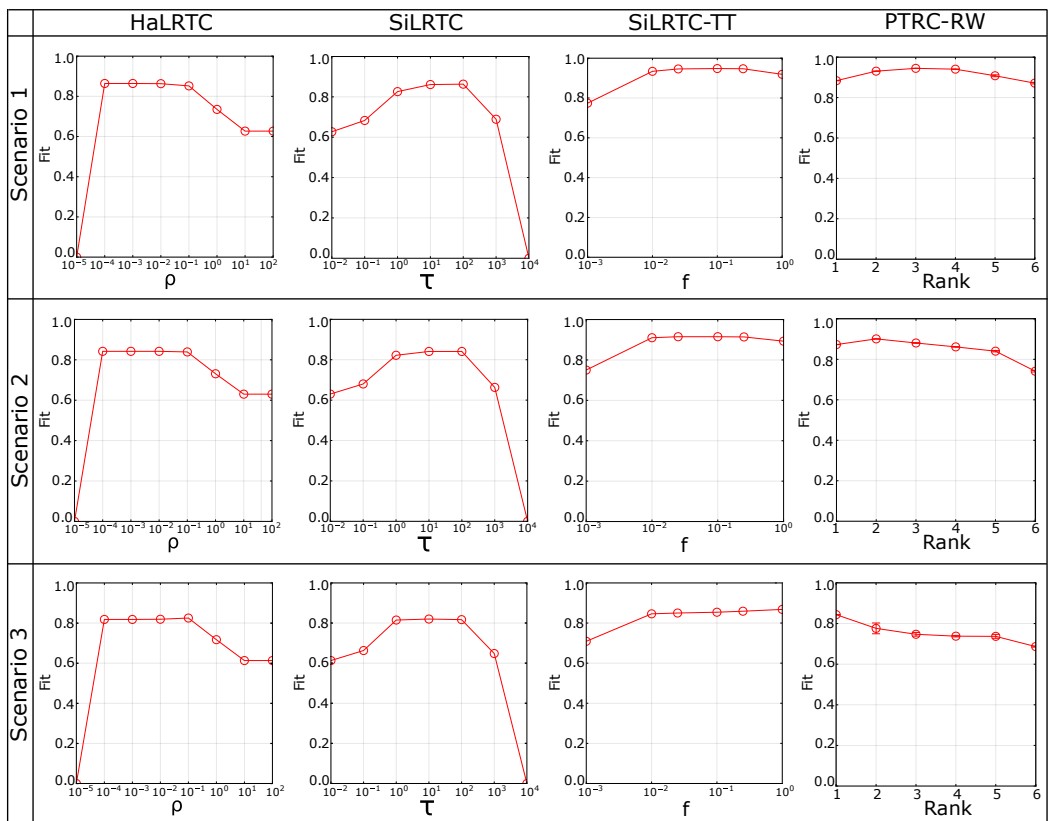

Figure 14: Hyper-parameter dependencies for the performance of baselines in the experiment in Section 3.2.

## References for Supplementary Materials

[42] Amari, S. 1998. "Natural gradient works efficiently in learning." *Neural computation* 10(2):251–276.

[43] Cheng, Song, Lei Wang, Tao Xiang and Pan Zhang. 2019. "Tree tensor networks for generative modeling." *Physical Review B* 99(15):155131.

[44] Kim, Yong-Deok and Seungjin Choi. 2007. Nonnegative tucker decomposition. In *2007 IEEE Conference on Computer Vision and Pattern Recognition*. IEEE pp. 1–8.

[45] Kossaifi, Jean, Yannis Panagakis, Anima Anandkumar and Maja Pantic. 2019. "TensorLy: Tensor Learning in Python." *Journal of Machine Learning Research* 20(26):1–6.
**URL:** *http://jmlr.org/papers/v20/18-277.html*

[46] Lee, Daniel and H Sebastian Seung. 2000. Algorithms for non-negative matrix factorization. In *Advances in Neural Information Processing Systems 13*. Denver, United States: pp. 556–562.

[47] Malik, Osman Asif and Stephen Becker. 2021. A sampling-based method for tensor ring decomposition. In *International Conference on Machine Learning*. PMLR pp. 7400–7411.

[48] Murg, Valentin, Frank Verstraete, Reinhold Schneider, Peter R Nagy and O Legeza. 2015. "Tree tensor network state with variable tensor order: An efficient multireference method for strongly correlated systems." *Journal of Chemical Theory and Computation* 11(3):1027–1036.

[49] Pedregosa, F., G. Varoquaux, A. Gramfort, V. Michel, B. Thirion, O. Grisel, M. Blondel, P. Prettenhofer, R. Weiss, V. Dubourg, J. Vanderplas, A. Passos, D. Cournapeau, M. Brucher, M. Perrot and E. Duchesnay. 2011. "Scikit-learn: Machine Learning in Python." *Journal of Machine Learning Research* 12:2825–2830.

[50] Shi, Y-Y, L-M Duan and Guifre Vidal. 2006. "Classical simulation of quantum many-body systems with a tree tensor network." *Physical review a* 74(2):022320.

[51] Walczak, B. and D.L. Massart. 2001. "Dealing with missing data: Part I." *Chemometrics and Intelligent Laboratory Systems* 58(1):15–27.
**URL:** *https://www.sciencedirect.com/science/article/pii/S0169743901001319*

[52] Zhang, Sheng, Weihong Wang, James Ford and Fillia Makedon. 2006. Learning from incomplete ratings using non-negative matrix factorization. In *Proceedings of the 2006 SIAM International Conference on Data Mining*. SIAM Bethesda, United States: pp. 549–553.
