# OpenReview forum: "Many-body Approximation for Non-negative Tensors"
_NeurIPS.cc/2023/Conference — NeurIPS 2023 poster_

### Official Review · Reviewer_wHyr · 2023-07-02

**Soundness:** 2 fair
**Presentation:** 3 good
**Contribution:** 3 good
**Rating:** 6
**Confidence:** 3

**Summary:**

Authors propose a tensor decomposition approach based on the Legendre decomposition and convex optimization with natural gradient-based algorithm for non-negative tensors which is inspired by many-body interactions in physics. The tensor is interpreted as a probability measure over a multidimensional discrete space given by its indices, and the complexity is regulated by the order and map of interactions between variables (this makes it possible to avoid the difficult problem of choosing the optimal rank of the decomposition). Experimental results demonstrating the good properties in terms of accuracy and computation time of the proposed method in comparison with a number of alternative methods of low-rank approximations.

**Strengths:**

The proposed method looks novel and interesting, and it has benefits like a convex optimization formulation and a lack of rank parameters to tune. The work is well structured, written in a good style and easy to read.

**Weaknesses:**

1. From the presented numerical experiments, the actual advantages of the proposed approach over baselines are not quite clear to me. For the case of a dataset with graphic images, what is the actual quality of the low-rank approximation (it would be useful to give examples of reconstructed images)?

2. As far as I know, there are effective rank-adaptive methods of low-rank approximations (rank-adaptive TT-cross method, rank-adaptive tensor ring, etc.), see, e.g., "Adaptive rank selection for tensor ring decomposition" (2021). Will the proposed approach be superior to such modern methods?

3. I did not see estimates of the computational complexity of the proposed method. What dimensions limit the applicability of the approach, i.e., can we use it, for example, for the 100-dim tensors?

**Questions:**

1. [Line 23] "More recently, tensor networks [4] have been introduced...". I do not fully agree with this position. Tensor networks have been used for a very long time in the scientific community, including quantum applications (e.g., DMRG), and also the noted decompositions (CP, Tucker) represent a case of the tensor network. It seems to me that this paragraph should be substantially reformulated. It also seems appropriate to note such decompositions as MERA, PEPS, etc.

2. [Line 268] It might be worth refining the positioning of the graphics so that the section title (3.2) isn't at the end of the page.

3. [Line 330] "error[s] are..."

4. [Lines 349-350] "visualizing interactions between modes ...  to see activated interactions between modes". Perhaps this sentence should be reformulated to avoid repeating the phrase.

**Limitations:**

The method can only be applied to tensors with non-negative elements. There are also likely to be problems associated with high computational complexity when trying to apply this approach to significantly multidimensional data arrays.

---

> ### Author Rebuttal · Authors · 2023-08-08
>
> I appreciate your comments. We provide our point-by-point response to each of your comments in the following.
> ### In the "Weaknesses" section
>
> > 1. From the presented numerical experiments, the actual advantages of the proposed approach over baselines are not quite clear to me.
>
> Many-body approximation has two significant practical advantages over baselines, the global convergence and intuitive model design:
> - Many-body approximation always converges to the globally optimal solution as described in lines 112-114 and Section A. Thus we do not have to repeat algorithms to find better initial values. In contrast, presented results in Section 3-3 for baselines are best results after multiple trials with random initial values (see line 326).
> - Many-body approximation has an advantage in terms of model selection. For example, it does not require hyperparameter tuning as described in Figure 14 and Section E.2, while it still has comparable recovery fit scores compared to baselines. This can be a significant advantage of the proposed method in practice. In traditional low-rank approximation, hyper-parameter tuning based on prior knowledge is difficult because its semantics is unclear. In contrast, the interactions in many-body approximation can be tuned easily by considering the expected association among modes of a given tensor (see more discussion in Sections 3.1 and C.2 in Appendix).
>
> > 1. For the case of a dataset with graphic images, what is the actual quality of the low-rank approximation?
>
> Please refer to the global response for the reconstruction images. Also, Figure 11 in Appendix shows that the proposal has the better reconstruction quality for the traffic dataset compared to two traditional low-rank approximations, non-negative Tucker decomposition and non-negative tensor-train decomposition.
>
> > 2. Will the proposed approach be superior to rank-adaptive methods?
>
> We would like to argue that our proposal has two significant advantages over rank-adaptive methods: convexity and intuitive interaction tuning.
>
> Rank-adaptive methods typically repeat low-rank approximation so that the reconstruction error becomes smaller than a threshold. This strategy requires a threshold as an additional hyperparameter and more computational cost, as shown in Table III in [a]. In addition, since it is non-convex optimization, each iteration for low-rank approximation cannot find the global optimal due to the initial value dependency, as described in Section IV in [a]. Moreover, interpretation or understanding of the meaning of the obtained rank is difficult as it is not on the original feature space. In contrast, since our proposal is based on convex optimization, there is no initial value dependency; thus, we do not have to run the algorithm multiple times with random initial values to find a better approximation. Furthermore, as described in Sections 3.1, 3.2, and C.1, the interaction is more interpretable than the traditional rank.
>
> [a] Sedighin, Farnaz, Andrzej Cichocki, and Anh-Huy Phan. "Adaptive rank selection for tensor ring decomposition." IEEE Journal of Selected Topics in Signal Processing 15.3 (2021): 454-463.
>
> > 3. I did not see estimates of the computational complexity of the proposed method.
>
> As described in line 117, our method is based on the natural gradient, and its computational complexity is cubic with respect to the number of parameters to be optimized. For example, the number of parameters for cyclic two-body approximation is described in equation (12). Also, we provide a computational speed comparison in Figure 10 in Appendix.
>
> > 3. What dimensions limit the applicability of the approach, i.e., can we use it, for example, for the 100-dim tensors?
>
> It depends on not the dimensionality but the number of non-zero natural parameters. Please also refer to the discussion about computational complexity in line 117. For huge-size tensors and much higher-order interactions, we need to consider efficient variations of the natural gradient, which can be our future work.
>
> ### In the "Questions" section
>
> > 1. Tensor networks have been used for a very long time in the scientific community, including quantum applications (e.g., DMRG), and also the noted decompositions (CP, Tucker) represent a case of the tensor network. It seems to me that this paragraph should be substantially reformulated. It also seems appropriate to note such decompositions as MERA, PEPS, etc.
>
> Thank you for pointing it out. We agree that tensor networks have a long history and have been used to evaluate wave functions and partition functions for physics. What we wanted to say in that paragraph is that tensor networks, introduced in physics, have recently been used in the machine learning community as well. To avoid misleads, we will revise that paragraph as follows:
> More recently, tensor networks, initially introduced in physics, have become popular in the machine learning community because it helps intuitive and flexible model design, including tensor train decomposition [24], tensor ring decomposition [34], and tensor tree decomposition [2]. Nowadays, traditional tensor structures for physics, such as MERA [b] and PEPS [c], are also used in machine learning.
>
> [b] 10.1088/2632-2153/abffe8 [c] 10.1103/PhysRevB.103.125117
>
> We will also addressee issues 2, 3, and 4 in the camera-ready version. We appreciate your careful reading.
>
> ### In the "Limitations" section
>
> > There are also likely to be problems associated with high computational complexity when trying to apply this approach to significantly multidimensional data arrays.
>
> Although this is a general problem of tensor factorizations, we would like to clarify that, as you can see in Figure 10 in Appendix, our methods work on tensors whose sizes are ten million scales, 20×20×20×20×20×20 = 64,000,000, with Intel Xeon CPU Gold 5218 and 128GB of memory (See Section E). Moreover, we have provided a way to handle larger tensors in Section B-2 in Appendix.

---

> > ### Comment · Reviewer_wHyr · 2023-08-13
> >
> > Thank you for your detailed response and the accompanying graphic images.
> >
> > 1. I do not fully agree with your statement `the interaction is more interpretable than the traditional rank`. For example, in Lines 292-295 you repeatedly refer to the intuition of interactions' choice, but such intuition is possible in a significantly limited number of practical problems.
> >
> > 2. With regard to computational complexity, I meant the general form of the expected dependence on the problem's dimension (the order of the tensor in terms of your work) and the number of elements in each mode. For example, for tensor ring and tensor train decompositions, the complexity is usually positioned as linear in these quantities (subject to limited rank). Your approach, as far as I understand, has a significantly higher complexity, however, for the case of the dimensions considered in your work, this is not a problem.
> >
> > 3. I noticed that in section 3.3 in paragraph "Synthetic data" you missed the link to Figure from the application, I advise you to add it to the final text.
> >
> > However, these comments are not fundamental. The approach you suggested seems interesting to the scientific community, and you provided relevant clarifications to my initial comments in the review, so I'm raising my rating from 5 to 6.

---

> > > ### Author Response · Authors · 2023-08-17
> > >
> > > We appreciate your careful reading of our response and acknowledging our contributions.
> > > We will revise our text according to your third suggestion in the camera-ready version.

---

### Official Review · Reviewer_gp7Q · 2023-07-06

**Soundness:** 3 good
**Presentation:** 3 good
**Contribution:** 4 excellent
**Rating:** 6
**Confidence:** 4

**Summary:**

The paper proposes a new type of tensor decomposition, based on considering an interaction network among the different modes of a tensor. Nonnegative tensors are considered and factorized based on a Legendre decomposition, which may be viewed as representing the tensor elements by an underlying probability distribution. The paper then proposes to model the tensor in terms of interactions, each of which couples two or more modes. These interactions may be represented by a tensor network, which contains no rank modes, e.g., T_{ijkl}=U_{ij}V_{jk}W_{kl}. By contrast, typical tensor decompositions, include auxiliary/contracted modes. Using the theory of probability distributions, the authors argue that the resulting optimization problem to factorize the tensor according to interaction network, is convex. A gradient based method is used for optimization. Connections are made to other types of tensor networks, demonstrating that the interaction network arises by imposing additional structure on such networks. Experimental results show that, on the datasets considered, the interaction/many-body approximation method can be competitive in terms of accuracy.


**Strengths:**

 * The proposed tensor decomposition type appears to be new and is interesting / has connections to probability theory and physics.
 * The use of probability theory to demonstrate convexity of the optimization problem is innovative, however, it appears to me that the assertion of convexity is incorrect, see weaknesses. [After reviewing the rebuttal, I follow at least the high-level motivation for why the approach is convex and revised my evaluationp]



**Weaknesses:**

 * The Legendre decomposition is defined vaguely in the paper and supplementary material, and as far as I can tell this is not a broadly known concept (discussed only in a recent line of ML papers). This makes the paper not self-contained, and makes it difficult to ensure correctness of the arguments. The optimization method and proof of convexity of the optimization problem are also proven hastily / with lack of good definition/theorem/lemma/proof structure, in particular restate the theorems being used from prior work (e.g., when discussing results based on linear dependence of parameters in the distribution).
 * My main concern is that from a tensor point of view, it seems impossible that decomposing a tensor into a many-body approximation for an arbitrary interaction network leads to a convex optimization problem. Here is a counterexample. Consider a sixth order tensor with 3 interaction terms, each acting on two independent modes, T_{ijklmn}=U_{ij}V_{kl}W_{mn}. If the dimensions of modes j, l, and n are 1, we've obtained a rank-1 approximation of the tensor T'_{ikm}=T_{i1k1m1}=U_{i1}V_{k1}W_{m1}. Its well known that rank-1 approximation, which also corresponds to the tensor eigenvalue problem, is nonconvex (often has many local minima). [After reviewing the rebuttal, I see that KL-divergence enables convexity based on prior results.]


**Questions:**

Please provide a counterargument to my assertion above that the optimization problem is in fact nonconvex. If I have somehow overlooked something on this, I would be much more supportive of the paper. If its indeed nonconvex, the paper needs to be corrected and re-evaluated.


**Limitations:**

None.

---

> ### Author Rebuttal · Authors · 2023-08-08
>
> We appreciate your careful reading and constructive comments. As described below, our proposal is ensured to be convex.
>
> > My main concern is that from a tensor point of view, it seems impossible that decomposing a tensor into a many-body approximation for an arbitrary interaction network leads to a convex optimization problem. Its well known that rank-1 approximation, which also corresponds to the tensor eigenvalue problem, is nonconvex.
>
> Our argument is correct because we use not the Frobenius norm but the KL divergence. As you demonstrate and we describe in Section 2.5, our proposal includes rank-1 approximation. However, instead of the Frobenius norm, our proposal optimizes the KL divergence from an empirical distribution (a given normalized non-negative tensor) onto a model manifold (a set of interaction-reduced tensors). When the cost function is defined with the Frobenius norm, the best rank-1 approximation is NP-hard, as seen in Equation (28) in [a], which is non-convex. In contrast, when the cost function is the KL divergence, it becomes a convex problem, as shown in [b]. To avoid the misleading, we will add the following sentence in line 243 in the camera-ready version:
> Finding the rank-1 tensor that minimizes the Frobenius norm from a given tensor is known to be an NP-hard problem [a]. However, it has been reported that finding the rank-1 tensor that minimizes the KL divergence from a given tensor is a convex problem [b].
>
> [a] DOI: 10.1145/2512329
> [b] DOI: 10.1109/ACSSC.2017.8335432
>
>
> > The Legendre decomposition is defined vaguely in the paper and supplementary material, and as far as I can tell this is not a broadly known concept. This makes the paper not self-contained, and makes it difficult to ensure correctness of the arguments.
>
> We agree that Legendre decomposition is not a broadly known concept. Thus, we will make a new section, “Formal definitions” in Appendix and add the following formal definition of Legendre decomposition in the camera-ready version.
>
> __Definition 1__: Legendre Decomposition [29]
>
> For a normalized non-negative tensor $\mathcal{P}$ and a set of indices $B$, many-body approximation of $\mathcal{P}$ is $\mathcal{Q}$ defined as
> $ \mathcal{Q} = argmin_{\mathcal{Q} \in \boldsymbol{\mathcal{B}}} D_{KL} (\mathcal{P},\mathcal{Q}) $
> where $\boldsymbol{\mathcal{B}} = \set{ \mathcal{Q} \mid \theta_{i_1,\dots,i_D} = 0 \ \ \mathrm{ if }\ \ (i_1,\dots,i_D)\notin B \text{ for } \theta\text{-parameters of } \mathcal{Q}\}$.
>
> In the same way, we will give a formal definition of many-body decomposition as __Definition 2__  in the camera-ready version.
>
> > The optimization method and proof of convexity of the optimization problem are also proven hastily / with lack of good definition/theorem/lemma/proof structure, in particular restate the theorems being used from prior work.
>
> We appreciate your constructive comments for readability.
>
> - For the optimization method :
> The optimization method is thoroughly described in Section 2.1.1, Section A, and Algorithm 1. This is the Newton method that optimizes KL divergence from a given distribution. We are happy to address that further if you could provide unclear points specifically.
>
> - For convexity of the proposal:
> We have discussed in Section A to carefully introduce the necessary definitions to prove the argument, i.e., geodesic, flatness, and projection. We will add the following theorem in Appendix B with proof to further clarify our claim.
>
> __Theorem 1__: The solution of many-body approximation is always unique, and the objective function of many-body approximation is convex.
>
> __Proof__:
> As we see in Definition 2, the objective function of the proposed many-body approximation is the KL divergence from an empirical distribution (a given normalized tensor) to a subspace $\boldsymbol{\mathcal{B}}$. We can immediately prove that a subspace $\boldsymbol{\mathcal{B}}$ is $e$-flat from the definition of the flatness [1, Chapter 2] for any $B$. The KL divergence minimization from a given distribution to $e$-flat space is called $m$-projection (see the second paragraph in Section A). The $m$-projection onto $e$-flat subspace is known to be convex and the destination is unique (see the third paragraph in Section A). Thus, the optimal solution of the many-body approximation is always unique, and the objective function is convex.
>
> --
>
> It is widely known that maximum likelihood estimation of ordinary Boltzmann machines without hidden variables is a convex problem. Since we optimize the KL divergence, the proposed many-body approximation is also maximum likelihood estimation that approximates a non-negative tensor, which is regarded as an empirical distribution, by an extended Boltzmann machine without hidden variables, as described in lines 147-153. As with ordinary Boltzmann machines, the maximum likelihood estimation of extended Boltzmann machines can be globally optimized. __Theorem 1__ is a general proof of this using Information Geometry. We will add this description in Appendix in the camera-ready version.
>
>
> > when discussing results based on linear dependence of parameters in the distribution.
>
> In the second paragraph titled “Flatness and projections” in Appendix A, the sentence “It is known that a subspace with linear constraints on natural parameters $\theta$ is $e$-flat” describes the general condition for $e$-flatness. However, to prove the convexity of the proposal, we need only a particular condition “A subspace with some of its natural parameters fixed at 0 is $e$-flat” and it is obvious from the definition of $e$-flatness. To simplify, we replace that sentence in Appendix A with this sentence. In the same way, in line 98, “it is known that a subspace with linear constraints on natural parameters $\theta$ is flat, called $e$-flat [1, Chapter 2]” will be replaced as “we can introduce a concept of flatness for a set of tensors” to increase the readability.

---

> > ### Comment · Reviewer_gp7Q · 2023-08-17
> >
> > Thanks for the comments and clarifications. I was not aware of the results regarding convexity of low rank tensor product approximation with use of KL divergence. Thanks also for pointing out the subsections where some things are clarified, I had overlooked a coupled of things in my prior pass. I have updated my review to a weak accept as I now see why correctness of convexity should follow.
> >
> > I refrained from a full accept, as I am concerned that, even if convexity follows from KL divergence, why does that make the proposed method better than alternative tensor decompositions with a KL divergence objective? As far as I can tell this is not addressed in the experimental comparison (only tensor ring is considered, but based on prior work at least CP or tensor train with KL divergence seem like plausible alternatives). The paper should also describe the prior results regarding convexity of low-rank tensor decomposition with KL divergence as an objective and survey for what decompositions this is applied / is possible. This point seems to be a core motivation of the whole approach/analysis.

---

> > > ### Author Response · Authors · 2023-08-18
> > >
> > > We appreciate your careful reading of our response and acknowledging our contributions.
> > >
> > > > Why does that make the proposed method better than alternative tensor decompositions with a KL divergence objective?
> > >
> > > We would like to clarify that low-rank tensor decompositions are non-convex except for the rank-1 case, even if the objective function is defined with the KL divergence. In contrast, our proposal is always convex, which makes our proposal more favorable than theirs in terms of the stability.
> > >
> > > > This is not addressed in the experimental comparison (only tensor ring is considered, but based on prior work at least CP or tensor train with KL divergence seem like plausible alternatives).
> > >
> > > In Figure 11 in Appendix, we have already provided comparisons of the proposal with non-negative Tucker decomposition (NTD) and non-negative tensor train decomposition (NTTF) when optimized with the Frobenius norm. We here further provide fit scores of NTD and Non-negative CP decomposition (CPAPR) using the same protocol in Figure 11(a), where these objective functions are defined with not the Frobenius norm but the KL-divergence from input tensor to the reconstructed tensor.
> > >
> > > #### NTD with the KL divergence
> > >
> > > | Rank | # Parameters | Fit score (worst) |  Fit score (best) |
> > > | ---- | ---- | ---- | ---- |
> > > | [1,1,1,1] | 69     | 0.835 | 0.835 |
> > > | [5,5,5,5] | 965   | 0.835 | 0.894 |
> > > | [6,6,6,6] | 1704 | 0.835 | 0.901 |
> > > | [11,11,11,11] | 15389| 0.835 | 0.916|
> > >
> > >
> > > #### Non-negative CP decomposition (CPAPR) with the KL divergence
> > >
> > > | Rank | # Parameters | Fit score (worst) |  Fit score (best) |
> > > | ---- | ---- | ---- | ---- |
> > > | 1 | 69     | 0.835 | 0.835 |
> > > | 3 | 204 | 0.835 | 0.877 |
> > > | 10 | 680 | 0.888 | 0.910 |
> > >
> > > (For larger ranks, the algorithm did not converge.)
> > >
> > > #### Many-body approximation
> > >
> > > | Body | # Parameters | Fit score |
> > > | ---- | ---- | ---- |
> > > | One | 64 | 0.835 |
> > > | Cyclic-two | 1052| 0.907 |
> > > | Two | 1418| 0.917 |
> > > | Three | 11762| 0.973 |
> > >
> > > Hyperparameters for CPAPR are set to default values of the TensorLy implementation, and those for NTD are described in Section E.2.
> > >
> > > Please note that NTD and CPAPR are convex only if their ranks are [1,1,1,1] and 1, respectively, and hence the worst and the best scores are the same in this case.
> > >
> > >
> > > > The paper should also describe the prior results regarding convexity of low-rank tensor decomposition with KL divergence as an objective and survey for what decompositions this is applied / is possible.
> > >
> > > The low-rank tensor decomposition with the KL divergence is non-convex except for the rank-1 case. We will add this description in the camera-ready version.

---

> > > > ### Comment · Reviewer_gp7Q · 2023-08-18
> > > >
> > > > In the KL divergence is not NP hard reference (10.1109/ACSSC.2017.8335432), Section IV introduces a convex objective for CP decomposition based on KL divergence. They do make use of inequalities to get it, so I still believe your statement regarding nonconvexity for other decompositions, but more mathematical justification or references seem needed to make/clarify this claim.
> > > >
> > > > Thanks for providing a comparison to CP/Tucker with KL divergence. The manybody approximation method seems to generally obtain reasonable results in experiment. Though in this comparison, its not clear to me what method CPAPR is. CP decomposition is perhaps the most natural choice for nonnegative tensors and there is a good bit of prior work on it focusing on nonnegativity and generalized loss functions. A comparison to library like PLANC would be more interesting than TensorLy. For strictly positive data, another effective method is to just take the logarithm of the data and minimize Frobenius norm relative to that. But I don't view addition of such experimental comparisons as necessary for publication.

---

> > > > > ### Author Response · Authors · 2023-08-20
> > > > >
> > > > > We appreciate your constructive comments and valuable feedback.
> > > > >
> > > > >
> > > > > > more mathematical justification or references seem needed to make/clarify this claim.
> > > > >
> > > > > To support this claim, we would like to list known results about non-convexity with their references in the following.
> > > > > - The objective function with the KL divergence to approximate a given tensor by a CP formatted tensor is non-convex. This is described in the last paragraph in Section 1.1 in [c] and Equation (2.5) in [d] assuming $m_i$ has low-CP-rank structure.
> > > > > - The objective function with the KL divergence to approximate a given tensor by a Tucker formatted tensor is non-convex with respect to all factors. This is described in the first paragraph in Section 3.1 in [e]. The beta-divergence used in the literature is a generalization of the KL divergence.
> > > > >
> > > > > [c] DOI: 10.1080/10556788.2015.1009977
> > > > > [d] DOI: 10.2172/1865529
> > > > > [e] Axel Marmoret, Florian Voorwinden, Valentin Leplat, Jérémy E Cohen, Frédéric Bimbot. "Nonnegative Tucker Decomposition with Beta-divergence for Music Structure Analysis of audio signals." GRETSI 2022: XXVIIIe Colloque. 2022.
> > > > >
> > > > > > its not clear to me what method CPAPR is.
> > > > >
> > > > > We apologize for the missing reference of CPAPR. It is CP decomposition that minimizes the KL divergence instead of the Frobenius norm, which was introduced in [f]. Since both CPAPR and many-body approximation assumes non-negativity and objective functions with KL-divergence, CPAPR is a natural choice of a baseline.
> > > > >
> > > > > Please note that, in [f], although Poisson distribution for input data is discussed to motivate the choice of the KL divergence, technically we do not need such a distribution to apply CPAPR.
> > > > >
> > > > > [f] DOI: 10.1137/110859063
> > > > >
> > > > > > A comparison to library like PLANC would be more interesting than TensorLy. For strictly positive data, another effective method is to just take the logarithm of the data and minimize Frobenius norm relative to that. But I don't view addition of such experimental comparisons as necessary for publication.
> > > > >
> > > > > Thank you for your valuable suggestions about heuristics and the useful library. We will consider them in our future work.

---

### Official Review · Reviewer_oVis · 2023-07-08

**Soundness:** 3 good
**Presentation:** 3 good
**Contribution:** 3 good
**Rating:** 7
**Confidence:** 3

**Summary:**

This work introduces a novel approach to non-negative tensor decomposition, termed "many-body approximation," which specifically addresses the relationship among modes of tensors. It is formulated as a variant of Legendre decomposition and realized through globally minimizes the Kullback-Leibler divergence. Additionally, this work explores the connections between the proposed method and existing methods, using interaction representation. A series of experimental results substantiate the superiority of the proposed method over other existing ones in dealing with tensor completion applications.

**Strengths:**

1. This paper is comprehensive and includes all the essential components. The structure is logical, and the design principle is effectively and adequately illustrated.
2. The proposed method is a novel rank-free method, which is particularly advantageous as identifying an appropriate low-rank structure can be highly challenging in some practical applications.
3. This paper visualizes the presence or absence of interactions between modes and further demonstrates the interpretability of interactions among all the modes.

**Weaknesses:**

1. The proposed decomposition is only applicable to non-negative tensors. This may restrict the method's effectiveness and generalizability when dealing with tensors that contain negative values.
2. Some theoretical analysis about the proposed "many-body approximation"method is missed.

**Questions:**

Is it possible to derive some theoretical results to support the merits of the proposed "many-body approximation" method in dealing with tensor completion and approximation problems?

**Limitations:**

See the Weakness\Questions above.

---

> ### Author Rebuttal · Authors · 2023-08-08
>
> We appreciate your positive feedback.
>
> > Some theoretical analysis about the proposed "many-body approximation" method is missed. Is it possible to derive some theoretical results to support the merits of the proposed "many-body approximation" in dealing with tensor completion and approximation problems?
>
> As described in lines 112-114 in the main text and the second paragraph titled “Flatness and projections”  in Section A, we have derived theoretical results of many-body approximation using information geometry, the uniqueness of the solution and the global convergence of the natural gradient method. These theoretical results directly lead to practical merits, that is, we do not have to run the algorithm multiple times with random initial values to find a better approximation.

---

> > ### Comment · Reviewer_oVis · 2023-08-19
> >
> > Thanks so much for your clarifications about the issue of theoretical analysis. I shall raise my rating from 6 to 7.

---

> > > ### Author Response · Authors · 2023-08-20
> > >
> > > Thank you for your response. We appreciate your acknowledgement of our contributions.

---

### Official Review · Reviewer_59t8 · 2023-07-11

**Soundness:** 3 good
**Presentation:** 2 fair
**Contribution:** 2 fair
**Rating:** 5
**Confidence:** 2

**Summary:**

The authors introduce a new, energy-model based approach to the decomposition of non-negative tensors. They compare their method to mainstay techniques.

**Strengths:**

The technique seems solid and reasonable, but how solid and reasonable is hard to assess (see below).

**Weaknesses:**

As presented, it's hard for me to determine how novel and distinct the presented method is. I work extensively with tensors, but as practical tools, so I am not fully versed in the various techniques and advantages for the various decomposition technique. For me to determine this, I would need a much more extensive literature review (perhaps adding a table) and to compare the new technique much more thoroughly and extensively to existing techniques (to highly the similarities, differences, advantages, and disadvantages).

**Questions:**

My questions all relate to the similarities and structures of the other techniques.

**Limitations:**

This is not an issue. The work presented by the authors is a general tool.

---

> ### Author Rebuttal · Authors · 2023-08-08
>
> Thank you for your comments about our novelty and impact.
>
> According to your suggestion, we have prepared a table comparing the proposed method with major tensor decomposition methods, the CP decomposition, Tucker decomposition, tensor train decomposition, and tensor ring decompositions. If there is any other information to be added, please let us know.
>
>
> | Models | Global Optimal | Uniqueness  of solution | Required parameter | Remarks |
> | ------------- | ------------- | ------------- | ------------- | ------------- |
> | CP decomp. | No | No  | CP-rank $r$  | NP-hardness, ill-posed |
> | Tucker decomp. | No | No  | Tucker rank $(r_1, \dots,r_D)$  | |
> | Tensor train decomp. | No | No  | Train rank  $(r_1, \dots,r_{D-1})$   | Optimal rank is imbalanced |
> | Tensor ring decomp. | No | No  | Ring rank $(r_1, \dots, r_D)$  | Cyclic structure |
> | Many-body Approx. | __Yes__ | __Yes__  | Interactions, or $m$ of $m$-body | Only for non-negative tensors|

---

> > ### Comment · Reviewer_59t8 · 2023-08-10
> > **Referee Response**
> >
> > The table is helpful, although it would be nice to have it compare other facets than just global optima and uniqueness of solution, and including additional discussion on the details would have been helpful as well. I'll keep my mildly positive review score.

---

> > > ### Author Response · Authors · 2023-08-17
> > >
> > > We appreciate your additional feedback on our response and positive review.
> > > Here we provide further discussion of tensor decomposition methods.
> > >
> > > In general, it is difficult to find out the low-rank structure of a given data set. Therefore, how to choose a low-rank decomposition model is hard to answer. This is why it is nontrivial to state the general advantages and disadvantages of each model itself despite various studies on optimization methods and regularization. To alleviate this difficulty in model selection and provide intuitive modeling, we have introduced the concept of body and formulated a decomposition that does not rely on ranks.
> > >
> > >
> > > We will add the above discussion in Appendix in the camera-ready version.

---

### Official Review · Reviewer_xQBv · 2023-08-05

**Soundness:** 3 good
**Presentation:** 3 good
**Contribution:** 2 fair
**Rating:** 6
**Confidence:** 3

**Summary:**

This paper proposes a novel many-body decomposition, based on Legendre decomposition, for non-negative tensor. Instead of specifying a rank parameter, the proposed method use energy-based model to the interactions among modes.

**Strengths:**

1. The paper is overall well-written and easy to follow. Especially the figures that explain the interactions.
2. The work is technically sound and aims at an interesting angle of tensor decomposition which is worth more attention from the research community.
3. The proposed method doesn't require the rank parameter which can save a headache of parameter tuning in many existing tensor analyses.

**Weaknesses:**

1. The novelty of the proposed method is limited given the existing Legendre decomposition.
2. The empirical advantages of the proposed methods are not sufficiently demonstrated.
3. The rank parameter is not necessarily required in traditional tensor methods, for example, tensor nuclear norm based approach.
4. The details of the proposed algorithm, LBTC, should be more emphasized in the main text.
5. How is the computational cost of the proposed approach, compared to the traditional tensor decompositions? There were many fast methods have been proposed for rank based decomposition, such as tensor CUR decompositions.
6. If I understand right, the proposed method is indeed a generalized rank-1 approximation.
7. "interactions between modes" -> "interactions among modes"


**Questions:**

See weakness section

**Limitations:**

1. As the authors already stated in the paper, the proposed method can only be applied to non-negative tensors.
2. Compared to rank-based tensor decomposition, the proposed method seems to require more expert knowledge about the application itself.

---

> ### Author Rebuttal · Authors · 2023-08-08
>
> We appreciate your positive feedback. We provide our point-by-point response to each of your comments in the following.
>
> ### In the "Weaknesses" section
>
>
> > 1. The novelty of the proposed method is limited given the existing Legendre decomposition.
>
>
> Although our proposal is based on Legendre decomposition, we have the following nontrivial novel contributions:
>
> - As described in lines 48-49, Legendre decomposition does not provide factorization, while the proposed many-body approximation factorizes tensors into a product form that enables us to observe components of the data. For example, as seen in Figure 3, our proposal factorizes the COIL image dataset into two components, the color tendency of each object (e) and the shape of each object (f). We also provide another example with the traffic dataset in Figure 13 in Appendix.
>
> - As described in lines 44-45, we have revealed that the $n$-body parameters control interaction among modes. Such a tight connection between natural parameters and interaction between modes was not known in literature of Legendre decomposition.
>
> - We have proved a theoretical connection between Legendre decomposition and low-rank approximation (Section 2.4).
>
>
> > 2. The rank parameter is not necessarily required in traditional tensor methods, for example, tensor nuclear norm based approach.
>
> Thank you for pointing this out. You are right that nuclear norm-based methods do not require the rank parameter. We will add the following text in line 32:
> Although there is an attempt to avoid rank tuning by approximating it by trace norms [2], it also requires another hyperparameter, the weights of the unfolding tensor, hence such an approach does not fundamentally solve the problem.
>
> --
>
> Please note that nuclear norm-based methods require other hyperparameters instead of ranks. In our experiments, the baseline methods SiLRTC and SiLRTC-TT in Section 3.2 are typical nuclear norm-based methods, and their performance depends on hyperparameters, as explained in Section E.2 and Figure 14. These dependencies were also noted in their original papers (for example please refer to Section VI. A in [2]).
>
> [2] Bengua, J. A., Phien, H. N., Tuan, H. D., and Do, M. N. (2017). Efficient tensor completion for color image and video recovery: Low-rank tensor train. IEEE Transactions on Image Processing, 26(5):2466–2479
>
>
> > 3. The empirical advantages of the proposed methods are not sufficiently demonstrated.
>
> We have demonstrated the empirical advantages of the proposed method in terms of hyperparameter tuning and model selection in Sections 3.2 and C.2, initial value dependency in Figure 11 in Appendix, and tensor completion in Section 3.2, which can be significant merits in practical situations.
>
> > 4. The details of the proposed algorithm, LBTC, should be more emphasized in the main text.
>
> Thank you for your suggestion. In the camera-ready version, we will add a new section 2.6 to describe LBTC, to which we put lines 269-280 and Algorithm 2.
>
> > 5. How is the computational cost of the proposed approach, compared to the traditional tensor decompositions? There were many fast methods have been proposed for rank based decomposition, such as tensor CUR decompositions.
>
> As described in line 117, our method is based on the natural gradient, and its computational complexity is cubic with respect to the number of parameters to be optimized, that is, the total complexity is $O(\gamma |B|^3)$, where $\gamma$ is the number of iterations. Due to the second-order convergence property of Newton's method, the iteration number $\gamma$ is usually small (Section C.3 in Appendix). For many low-rank decomposition methods, the computational complexity is cubic with respect to the rank. For example, the computational cost of tensor train decomposition of a $d$-order tensor of size $n×n× \dots ×n$ is $O(dnr^3)$, where the rank is $(r,r,…,r)$ [a].
>
> Please note that your suggested acceleration techniques based on SVD, such as CUR decomposition, cannot directly incorporate the non-negative condition in decomposition. As seen in Figure 10 in Appendix, we have also compared runtime of cyclic two-body approximation methods, NTR-MM and NTR-lraMM, that are known as faster methods than traditional tensor ring decomposition [32].
>
> [a] Oseledets, Ivan V. "Tensor-train decomposition." SIAM Journal on Scientific Computing 33.5 (2011): 2295-2317.
> [32] Yu, Y., Xie, K., Yu, J., Jiang, Q., and Xie, S. (2021b). Fast nonnegative tensor ring decomposition based on the modulus method and low-rank approximation. Science China Technological Sciences, 64(9):1843–1853.
>
>
> > 6. If I understand right, the proposed method is indeed a generalized rank-1 approximation.
>
> Your understanding is correct. We have discussed that point in Section 2.5. We have shown in this paper the significant potential of a generalization of rank-1 approximation by formulating it as a many-body approximation, which has comparable results with traditional non-convex low-rank approximation.
>
> > 7. "interactions between modes" -> "interactions among modes"
>
> Thank you for pointing out that. We will revise that in the camera-ready version.
>
> ### In the "Limitations" section
>
> > Compared to rank-based tensor decomposition, the proposed method seems to require more expert knowledge about the application itself
>
> In low-rank approximation, we often gradually increase the rank to enlarge model capability. In the same way, if we have no prior or domain knowledge about a tensor, we can gradually increase $m$ of $m$-body approximation and get more accurate results. Hence, even when the user knows nothing about the data, the proposed method is also easy to use.

---

> > ### Comment · Reviewer_xQBv · 2023-08-10
> >
> > Thank you for the authors' response. It has addressed most of my concerns. I will keep my positive score.

---

> > > ### Author Response · Authors · 2023-08-17
> > >
> > > We appreciate your careful reading of our response and acknowledging our contributions.

---

### Author Rebuttal · Authors · 2023-08-09

We thank all the reviewers for their insightful feedback and constructive suggestions. We responded to individual reviewers and look forward to further discussions.

Also, to address the concern raised by Reviewer wHyr, we submit a PDF file of reconstructed images of the COIL dataset to compare proposal and low-rank approximations, non-negative Tucker decomposition (NTD) and non-negative tensor train decomposition (NTT).

---

### Decision · Program_Chairs · 2023-09-21

**Decision:**

Accept (poster)

**Comment:**

Overall the reviewers all appreciate the idea and a decomposition but all flagged issues in clarity. Given the reviewers all vote for acceptance and agree this will be of interest to the community, I recommend acceptance. I strongly suggest the authors update the papers according to the discussion with the reviewer and revisit the paper for clarity.